# Can I Have Your Order? Monte-Carlo Tree Search for Slot Filling Ordering in Diffusion Language Models

**Joshua Ong Jun Leang** [1 2]   **Yu Zhao** [2]   **Mihaela Cătălina Stoian** [1]
**Wenda Li** [2]   **Shay B. Cohen** [2]   **Eleonora Giunchiglia** [1]

## Abstract

While plan-and-infill decoding in Masked Diffusion Models (MDMs) shows promise for mathematical and code reasoning, performance remains highly sensitive to slot infilling order, often yielding substantial output variance. We introduce MCDIFFUSE, a framework that formulates slot selection as decision making and optimises infilling orders through Monte Carlo Tree Search (MCTS). MCDIFFUSE uses look-ahead simulations to evaluate partial completions before commitment, systematically exploring the combinatorial space of generation orders. Experiments show an average improvement of 3.2% over autoregressive baselines and 8.0% over baseline plan-and-infill, with notable gains of 19.5% on MBPP and 4.9% on MATH500. Our analysis reveals that while MCDIFFUSE predominantly follows sequential ordering, incorporating non-sequential generation is essential for maximising performance. We observe that larger exploration constants, rather than increased simulations, are necessary to overcome model confidence biases and discover effective orderings. These findings establish MCTS-based planning as an effective approach for enhancing generation quality in MDMs.

## 1. Introduction

While autoregressive models (ARMs; DeepSeek-AI 2024; Team 2025) have achieved remarkable progress across a wide range of reasoning tasks (Wei et al., 2022; Leang et al., 2025a; Lyu et al., 2023), their inference is fundamentally limited by the sequential constraint of autoregressive decoding. Masked Diffusion Models (MDMs; Bie et al. 2026; Ye et al. 2025) have been proposed to overcome the limitations of current ARMs through an iterative denoising process with no fixed generation order. MDMs enable non-sequential order decoding by assuming conditional independence among target tokens and offer the potential to discover generation orders beyond rigid left-to-right trajectories (Li et al., 2023; Kim et al., 2025; Fu et al., 2025). While MDMs offer inference efficiency, they often underperform compared to ARMs (Nie et al., 2025). One critical reason is that such generation can lead to interdependent tokens generated simultaneously without mutual conditioning, making output quality sensitive to the generation order (Li et al., 2025).

Recently, the "plan-and-infill" framework (Li et al., 2025) has emerged as a promising approach for optimising generation orders in MDMs, where each denoising iteration comprises two steps: *a planning phase* that selects a sub-sequence (slot) from the set of currently masked sub-sequences, and *an infilling phase* that generates tokens within this selected slot autoregressively. While this framework alleviates the difficulty of generating tokens with strong local, inner-slot dependency (Li et al., 2025), the inter-slot generation order remains critical to output quality, as errors in ordering can induce unmodelled dependencies that propagate across iterations and undermine global coherence. Identifying such orderings is challenging due to the vast combinatorial space of possible slot permutations. Consequently, heuristic planners, such as confidence-based selection used in current MDMs (Ye et al., 2025; Schuster et al., 2022), often fail to account for long-range dependencies, leading to performance degradation.

To overcome the difficulty of reliably selecting slots for infilling (i.e., *the planning phase*), we introduce **M**onte **C**arlo **Diffu**sion **Se**arch (MCDIFFUSE), a training-free framework that formulates slot selection as a decision-making problem. Rather than seeking a provably optimal schedule (Figure 1), MCDIFFUSE performs targeted lookahead over the combinatorial space of slot orderings using Monte Carlo

---

[1]Imperial College London [2]University of Edinburgh. Correspondence to: Joshua Ong Jun Leang <j.ong25@imperial.ac.uk>, Eleonora Giunchiglia <e.giunchiglia@imperial.ac.uk>.

*Proceedings of the 43rd International Conference on Machine Learning*, Seoul, South Korea. PMLR 306, 2026. Copyright 2026 by the author(s).

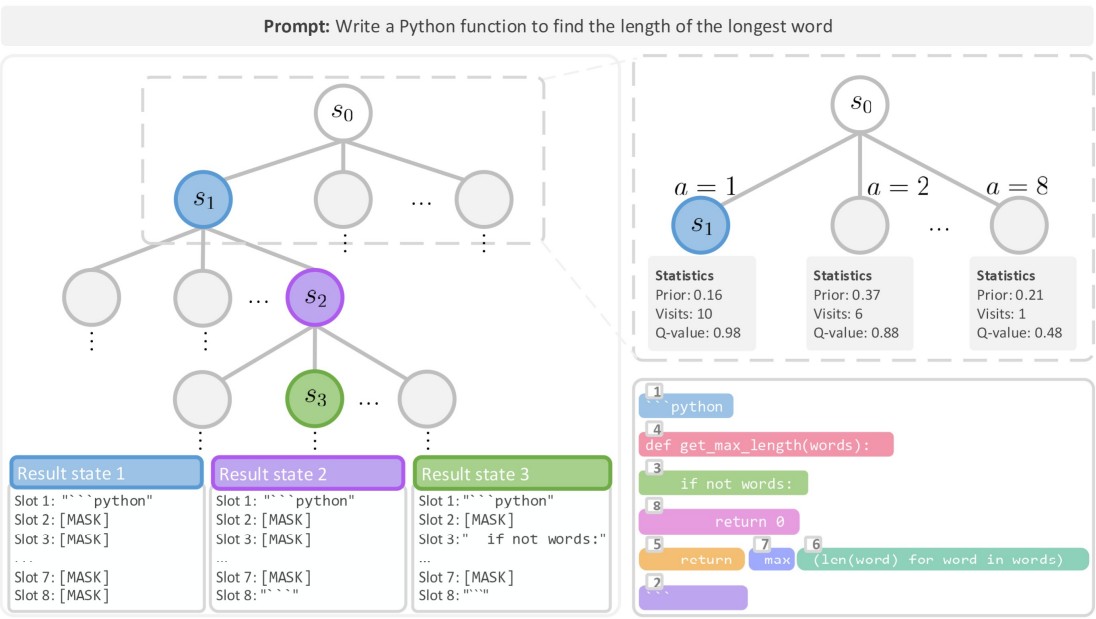

*Figure 1.* Overview of MCDIFFUSE. We formulate slot selection as a sequential decision-making process optimised via Monte Carlo Tree Search. As illustrated in the *Statistics* box, the model's greedy prior ($P(a = 2 \mid s_0) = 0.37$) favours immediately generating the function definition (i.e., slot 2: "`def get_max_length(words):`"). However, through look-ahead simulations, the search algorithm discovers that starting with the syntax declaration (i.e., slot 1: "` ```python`") yields a higher long-term Q-value (i.e., $Q(s_0, a = 1) = 0.98$ for slot 1 vs. $Q(s_0, a = 2) = 0.88$ for slot 2), allowing the model to override the biased local prior and ensure global coherence.

Tree Search (MCTS; Kocsis & Szepesvári 2006) with prior-guided expansion (Silver et al., 2017). In particular, the search integrates the model's intrinsic confidence scores as prior probabilities and uses a hybrid reward mechanism combining immediate denoising quality with rollout-based estimation of long-term trajectory coherence. By simulating future generation steps, MCDIFFUSE identifies slot orderings that reduce error propagation across iterations, thereby improving generation quality.

We evaluate MCDIFFUSE across six reasoning benchmarks, where it consistently outperforms current MDMs and the state-of-the-art "plan-and-infill" method, ReFusion (Li et al., 2025). Specifically, MCDIFFUSE achieves absolute accuracy gains of 19.45% on MBPP (Austin et al., 2021b) and 4.9% on MATH500 (Hendrycks et al., 2021). MCDIFFUSE narrows the performance gap between diffusion and autoregressive models, matching or exceeding autoregressive performance on five out of six reasoning benchmarks under identical experimental conditions. These results demonstrate the critical importance of strategic slot planning in MDMs and confirm the effectiveness of MCDIFFUSE's trajectory exploration approach.

We further provide a comprehensive analysis revealing two key insights into MCDIFFUSE's success. (1) While MCDIFFUSE predominantly adheres to sequential (left-to-right) generation, it strategically deviates to non-sequential orderings for a critical subset of samples, achieving higher

accuracy on these cases and demonstrating that *incorporating non-sequential generation is essential for maximising performance on reasoning tasks*. (2) Unlike traditional MCTS (Guan et al., 2025; Yang et al., 2025) applied to ARMs, our approach prioritises exploration over simulation depth. We find that an increasing number of simulations does not consistently improve performance. Instead, a large exploration constant is necessary to overcome the model's confidence priors and discover effective slot orderings, indicating that the primary challenge in slot planning lies in initiating search down low-prior branches to ensure sufficient exploration breadth, escaping bias towards locally confident but globally incoherent initial priors (Section 6).

In summary, our contributions are: 1) we propose MCDIFFUSE, a training-free framework that tailors MCTS specifically for searching slot orderings in MDMs, effectively closing the performance gap with ARMs; 2) we provide empirical analysis revealing that strategic non-sequential generation is critical to performance, validating MCTS as an effective approach for slot orderings; 3) we show that exploration breadth is critical to slot planning, requiring a large exploration constant to escape biased confidence priors and discover better orderings.

## 2. Background

**Markov decision process.** Sequential decision-making problems are typically modelled as a *Markov Decision Pro-*

*cess* (MDP; Bellman 1957). Formally, an MDP is a tuple $\langle \mathcal{S}, \mathcal{A}, \mathcal{T}, \mathcal{R} \rangle$, where $\mathcal{S}$ is the *state space* representing all of the possible configurations of the environment, and $\mathcal{A}$ is the *action space*. At every time step $t$, an agent in state $s_t \in \mathcal{S}$ selects an action $a \in \mathcal{A}$, upon which the environment transitions to a successor state $s_{t+1} \in \mathcal{S}$ according to the *transition probability function* $\mathcal{T}(s_{t+1} \mid s_t, a)$. The agent receives a reward specified by the *reward function* $\mathcal{R}$, and the objective is to learn a policy that maximises the expected cumulative reward over time.

**Monte Carlo tree search.** *Monte Carlo Tree Search* (MCTS) is a heuristic search algorithm that approximates optimal policies in MDPs by combining tree search with Monte Carlo simulation to balance exploration and exploitation (Coulom, 2006). In the tree, each node represents a state $s \in \mathcal{S}$ and each edge represents an action $a \in \mathcal{A}$. MCTS proceeds through repeated simulations, each consisting of four phases: (i) selection, (ii) expansion, (iii) simulation, and (iv) backpropagation.

During (i) *selection*, the search starts from the root node corresponding the current state and traverses the tree by selecting child nodes according to a bandit-based policy. Upon reaching a leaf node, the (ii) *expansion* step instantiates child nodes corresponding to previously unexplored actions available at that state. One of these newly created children is then selected for evaluation. To estimate its value, a (iii) *simulation* (or *rollout*) is performed by sampling a trajectory from the current state until a terminal condition is met, often using a user-defined lightweight policy to guide action selection. Finally, during (iv) *backpropagation*, the cumulative reward resulting from the simulation is propagated backward up the tree. After backpropagation, a new simulation begins again from the root. This process repeats until a predefined computational budget is exhausted. A common policy choice used during the selection phase is the Upper Confidence Bounds for Trees (UCT; Kocsis & Szepesvári 2006):

$$\mathrm{UCT}(s, a) = Q(s, a) + c_{\mathrm{uct}} \sqrt{\frac{\ln N_s}{N_s^a}}, \qquad (1)$$

where $Q(s, a)$ is the empirical mean return over all rollouts in which action $a$ was selected from state $s$, $N_s$ is the number of visits of state $s$, and $N_s^a$ is the number of times action $a$ was selected from $s$. Visit counts are initialised to zero upon node instantiation and are updated at each iteration of the algorithm during backpropagation. The constant $c_{\mathrm{uct}} > 0$ controls the exploration–exploitation trade-off, i.e., a larger $c_{\mathrm{uct}}$ value corresponds to higher exploration.

**Neural-guided MCTS.** In high-dimensional action spaces, modern MCTS variants such as AlphaZero (Silver et al., 2017) replace stochastic rollouts with learned

value functions and incorporate neural policy priors to guide exploration. These methods typically employ the *Predictor-Upper Confidence Tree* (PUCT) algorithm which selects children using the criterion defined as:

$$\mathrm{PUCT}(s, a) = \underbrace{Q(s, a)}_{\text{Exploitation term}} + \underbrace{c \cdot P(a \mid s) \cdot \frac{\sqrt{N_s}}{1 + N_s^a}}_{\text{Exploration term}}, \quad (2)$$

where $P(a \mid s)$ denotes the prior probability of selecting action $a$ in state $s$, as predicted by a neural policy $\pi_\theta$. The policy prior might be learnt independently (e.g., via reinforcement or imitation learning (Silver et al., 2017; Granter et al., 2017)) or jointly with MCTS by training the network to match the search-induced action distribution (Schrittwieser et al., 2020). The exploration term biases early search toward actions with high prior probability while ensuring that the influence of the prior diminishes as visit counts increase. This mechanism effectively reduces the search effort by focusing exploration on semantically plausible actions and is particularly well suited for inference-time optimisation. The definition of $Q(s, a)$, $N_s$, $N_s^a$ and $c$ is as in Equation 1.

## 3. McDiffuSE

In this section, we formulate the slot planning as an MDP (Section 3.1), and introduce our MCTS-based slot planning approach for determining infilling orders (Section 3.2).

### 3.1. Problem Formulation

Suppose we have a generative model with parameters $\theta$ that produces text as a sequence $\mathbf{x} = (\mathbf{x}^1, \ldots, \mathbf{x}^K)$ of $K$ contiguous and disjoint *slots*, i.e., strings each with $L$ tokens. While tokens within each slot are generated autoregressively, slots themselves may be generated in an arbitrary order.

Our goal is to determine an ordering of the slots that reflects how the generation of one slot influences the others. We represent such ordering as a permutation $(\sigma_1, \sigma_2, \ldots, \sigma_K)$, where $\sigma_t$ is the index of the slot generated at step $t$, with $t = 1, \ldots, K$. We formulate this optimisation problem as a deterministic MDP $\langle \mathcal{S}, \mathcal{A}, \mathcal{T}, \mathcal{R} \rangle$, whose components are defined below.

**State space ($\mathcal{S}$):** A state encodes which slots have been generated so far and the corresponding partially filled sequence. Formally, a state $s_t \in \mathcal{S}$ visited at step $t$ is defined as a pair $(\sigma_{:t}, \mathbf{x}_t)$, where $\sigma_{:t} = (\sigma_1, \ldots, \sigma_t)$ represents the prefix of $\sigma$ containing the indexes of the slots generated up to step $t$, while $\mathbf{x}_t = (\mathbf{x}_t^1, \ldots, \mathbf{x}_t^K)$ is the current (possibly partially filled in) text sequence. For any slot index $k \notin \{\sigma_1, \ldots, \sigma_t\}$, the corresponding slot $\mathbf{x}_t^k$ is populated with the special [MASK] token. The first state ($t = 0$) is initialised with $\sigma_{:0} = \emptyset$[1] and $\mathbf{x}_0 = $ [MASK]$^K$, where

---

[1] We use $\emptyset$ to denote an empty sequence.

$[\texttt{MASK}]^K$ indicates a sequence of $K$ $[\texttt{MASK}]$ tokens.

**Action space ($\mathcal{A}$):** An action corresponds to selecting the index of one of the masked slots. Hence, for every $a \in \mathcal{A}$, $a \in \{1, \ldots, K\}$. At step $t$, the admissible[2] action set will be $\mathcal{A}_t = \{k \mid k \in \{1, \ldots, K\} \setminus \{\sigma_1, \ldots, \sigma_t\}\}$.

*Example* 3.1. Consider Figure 1, where the following prompt is given *"Write a Python function to find the length of the longest word"*, and where the target response is partitioned into $K = 8$ slots (e.g., slot 1: "```python", slot 2: "def get_max_length(words):", etc.). The initial state $s_0$ has all slots masked. If slot 1 is selected ($a = 1$), then the successor state becomes $s_1$, where slot 1 is infilled (e.g., with "```python") and the other slots are still masked (see bottom left side of the Figure).  ◁

**Transition function ($\mathcal{T}$):** In our formulation the transition function is deterministic, and hence, given a state $s_t$ and an action $a = \sigma_{t+1}$, $\mathcal{T}(s_{t+1} \mid s_t, a)$ assigns probability 1 to a unique successor state. Since the slot chosen by the action $a$ is infilled via argmax decoding of the generative model, $s_{t+1} = (\sigma_{:t} \oplus \sigma_{t+1}, \mathbf{x}_{t+1})$, where $\oplus$ is used to indicate that we concatenate the index $\sigma_{t+1}$ to the sequence $\sigma_{:t}$, while the updated sequence $\mathbf{x}_{t+1}$ is identical to $\mathbf{x}_t$ except at position $\sigma_{t+1}$, in which the mask is replaced by the model's predicted tokens.

**Reward function ($\mathcal{R}$):** In the absence of ground-truth references during inference, we rely on the model's intrinsic confidence estimates to guide the tree search. Following recent work on confidence-based generation showing confidence correlates positively with downstream accuracy (Leang et al., 2025b; Nie et al., 2025; Prabhudesai et al., 2025; Damani et al., 2025), we define rewards based on the probability the model assigns to its own predictions. We define the reward as the *slot-level confidence*:

$$\mathcal{R}(s_t, a) = \frac{1}{L} \sum_{i=1}^{L} \mathbb{P}_\theta(\mathbf{x}_{t+1}^{\sigma_{t+1}}[i] \mid \mathbf{x}_t, \mathbf{x}_{t+1}^{\sigma_{t+1}}[<i], \text{Prompt}) \quad (3)$$

where $\mathbf{x}_{t+1}^{\sigma_{t+1}}[i]$ (resp. $\mathbf{x}_{t+1}^{\sigma_{t+1}}[< i]$) denotes the $i$-th token (resp. all tokens up to the $i$-th) of the slot at index $\sigma_{t+1}$, and $L$ is the slot size. At every step $t$, the reward $\mathcal{R}(s_t, a)$ quantifies the generation quality of the slot filled after taking action $a$, i.e., of the slot indexed by $\sigma_{t+1}$. Specifically, it is computed as the mean probability assigned by the generative model to all tokens in that slot. While confidence-based signals could, in principle, bias search toward locally easy-to-predict behaviours such as short continuations, our method is designed to mitigate this risk through two mechanisms. (1) The confidence is not used as a greedy decision rule, with the prior only initialising exploration and with the action selection being ultimately governed by PUCT;

(2) We ensure that the value function combines immediate confidence with rollout return.

## 3.2. Slot Planning via Monte Carlo Tree Search

To solve the MDP defined above without training auxiliary networks, we adapt MCTS to guide slot ordering by designing a rollout-based confidence estimator that captures future coherence beyond immediate slot-level confidence rather than relying on greedy local heuristics. We describe the four phases of the resulting MCTS procedure below.

**Selection.** Starting from the root node $s_0$, the algorithm recursively traverses the search tree by selecting actions according to the PUCT criterion in Equation (2) until an unexpanded leaf node is reached. For previously unvisited state–action pairs, the corresponding value estimates are initialised as $Q(s, a) = 0$.

*Example* 3.2. *(cont'd Example 3.1)* At the root state $s_0$, MCTS evaluates the admissible actions corresponding to the first slot to be filled. The prior $P(a \mid s_0)$ computed from the slot level confidence (as detailed in Equation (4)), favours generating the function definition (i.e, slot 2). However, the Q-values estimated via look-ahead simulations assign higher long-term value to starting with the syntax declaration (slot 1): $Q(s_0, 1) = 0.98$, $Q(s_0, 2) = 0.88$. Since PUCT balances the empirical value $Q(s_0, a)$ and the prior $P(a \mid s_0)$, the action $a = 1$ is selected despite the lower prior. This choice reflects the structural dependency in the program: generating the syntax declaration first induces a more coherent trajectory, as it conditions the remaining slots, enforces the overall code structure, and simplifies indentation handling, which is central to Python.  ◁

**Expansion.** When an unexpanded leaf node corresponding to state $s_t$ is reached, we expand the node by creating child nodes for all admissible actions $a \in \mathcal{A}_t$. For each newly created state-action pair $(s, a)$, visit counts $N_s$, $N_s^a$ and accumulated value $W(s, a)$ are initialised to zero. The prior over actions is obtained by a forward pass of the generative model conditioned on the current state $s_t$, from which we extract the slot-level confidence score $\mathcal{R}(s_t, a)$ for each unfilled slot. These scores are normalised to form the action prior:

$$P(a \mid s_t) = \frac{\mathcal{R}(s_t, a)}{\sum_{a' \in \mathcal{A}_t} \mathcal{R}(s_t, a')}. \quad (4)$$

Following expansion, a child node is selected according to the PUCT criterion to initiate the simulation phase.

*Example* 3.3. *(cont'd Example 3.2)* After selecting $a = 1$ at the root, the search reaches state $s_1$, where only one slot has been infilled. During the expansion step at $s_1$, child nodes are created for the remaining admissible actions $\{2, 3, \ldots, 8\}$. Conditioned on the filled slots at $s_1$, the

---

[2]An action is "admissible" at step $t$ if the corresponding slot has not yet been filled; $\mathcal{A}_t$ shrinks at each trajectory as slots are selected and $\mathcal{A}_t \neq \mathcal{A}$ for $t > 0$.

---

**Algorithm 1** Simulation Step

---

1: **Input:** Current state $s_t$, temperature $\tau$, mixing coefficient $\lambda$, slot-level confidence scores $\{\mathcal{R}(s_t, a)\}_{a \in \mathcal{A}_t}$
2: **Output:** Rollout estimate
3: $s_r \leftarrow s_t$  *// Copy state for lookahead*
4: $T \leftarrow 0, G \leftarrow 0$
5: **while** $s_r$ has unfilled slots **do**
6:    $\mathcal{A}_r \leftarrow \{$indices of unfilled slots in $s_r\}$
7:    *// Stochastic selection based on temperature*
8:    Compute probabilities $\pi_{\text{roll}}(a \mid s_r) \propto \exp(\mathcal{R}(s_r, a)/\tau)$ for all $a \in \mathcal{A}_r$
9:    Sample next slot action $\tilde{a} \sim \{\pi_{\text{roll}}(a \mid s_r)\}$
10:   $G \leftarrow G + \mathcal{R}(s_r, \tilde{a})$
11:   $s_r \leftarrow \mathcal{T}(s_r, \tilde{a})$ by filling slot $\tilde{a}$  *// Simulate infill*
12:   $T \leftarrow T + 1$
13: **end while**
14: $G \leftarrow G/T$ **if** $T > 0$ **else** $0$  *// Weighted combination*
15: **return** $G$

---

model induces a new prior distribution over the remaining admissible slots. ◁

**Simulation.** To evaluate an expanded leaf corresponding to state $s_t$, we estimate the value of taking action $a$ by combining the immediate slot-level reward with a stochastic rollout estimate:

$$V(s_t, a) = \lambda \cdot \mathcal{R}(s_t, a) + (1 - \lambda) \cdot \mathbb{E}_{\pi_{\text{roll}}}[G]. \quad (5)$$

The mixing coefficient $\lambda$ balances immediate local confidence against long-term trajectory quality. This balance is important as evaluating a leaf node solely based on the immediate confidence $\mathcal{R}(s_t, a)$ can be misleading: a slot that appears highly confident in isolation may constrain the model's predictions for remaining slots, leading to poor overall generation quality. The rollout term $G$ approximates the remaining return by continuing slot infilling from the successor state $s_{t+1} = \mathcal{T}(s_t, a)$ until a terminal state is reached. At each rollout step $r$, given a state $s_r$, we compute $\mathcal{R}(s_r, a)$ for all admissible actions $a \in \mathcal{A}_r$ and sample the next slot according to a temperature-scaled (according to $\tau$) softmax policy:

$$\pi_{\text{roll}}(a \mid s_r) = \frac{\exp(\mathcal{R}(s_r, a)/\tau)}{\sum_{a' \in \mathcal{A}_r} \exp(\mathcal{R}(s_r, a')/\tau)}. \quad (6)$$

The trajectory score is the average reward accumulated along the rollout,

$$G = \frac{1}{T} \sum_{i=1}^{T} \mathcal{R}(s_i, \tilde{a}_i), \quad (7)$$

where $(s_i, \tilde{a}_i)$ are the state–action pairs with $\tilde{a}_i \sim \{\pi_{\text{roll}}(a \mid s_i)\}$ encountered during the rollout and $T$ is the rollout length.

*Example* 3.4. *(cont'd Example 3.3)* From the newly expanded node, the rollout policy stochastically fills the remaining slots to estimate a trajectory value. For example,

it might sample the path: fill slot 3 (" if not words:") → slot 8 (closing tag "```") → slot 2 (function header "`def get_max_length(words):`"). ◁

We provide the pseudocode for the rollout process in Algorithm 1. Further, in Section B, we present a concrete numerical example of how the rollout values are calculated.

**Backpropagation.** After evaluating a leaf node, the estimated value $V(s, a)$ is recursively propagated back to the root along the traversal path. For each visited (state, action) pair $(s, a)$, we update the visit counts

$$N_s \leftarrow N_s + 1, \qquad N_s^a \leftarrow N_s^a + 1, \quad (8)$$

and accumulate the total value associated with the edge $(s, a)$ as
$$W(s, a) \leftarrow W(s, a) + V(s, a). \quad (9)$$

The empirical mean action value is then given by

$$Q(s, a) = \frac{W(s, a)}{N_s^a}, \quad (10)$$

which serves as the exploitation term in the PUCT selection rule.

After $N_{\text{sim}}$ simulations, we select the next action at the root of the current search tree, i.e., the current state $s_t$, using the robust child criterion

$$a_t^* = \arg\max_{a \in \mathcal{A}_t} N_{s_t}^a, \quad (11)$$

which chooses the action most frequently selected during search. While PUCT guides exploration during the selection phase of tree search, the final action selection relies solely on visit counts, eliminating the exploration bonus at decision time. We execute $a_t^*$ to transition to $s_{t+1}$ and MCTS again with $s_{t+1}$ as the new root state, repeating this process until all slots are filled.

*Example* 3.5. *(cont'd Example 3.4)* The high value obtained from constructing a syntactically valid Python function is propagated up the tree. As the last step of our MCTS-based algorithm, the root node updates its statistics for action $a = 1$ (including the mean value estimates and visit counts), reinforcing the decision to start with markdown Python code-fence declaration, rather than generating directly the code, yields the preferred generation path. Finally, after $N_{\text{sim}}$ simulations are completed, a single child of the root node is selected corresponding to the highest visit count. In the context of Figure 1, this results in selecting the blue node (corresponding to transitioning from state $s_0$ to $s_1$ through action $a = 1$), which now marks the syntax declaration as the next slot to be infilled. ◁

While Algorithm 1 details our core value estimation strategy (the rollout), the full algorithm for slot selection is provided in Section A. Further, in Section F, we provide a concrete, step-by-step walkthrough of the four MCTS phases of our approach, across multiple simulations.

| Model | GSM8K | MATH500 | ARC | GPQA-Diamond | MBPP | HumanEval | Average |
|---|---|---|---|---|---|---|---|
| *Autoregressive Models* | | | | | | | |
| Llama-3.1 8B | 83.48% $\pm 0.64\%$ | 35.90% $\pm 1.56\%$ | 83.32% $\pm 0.06\%$ | 31.14% $\pm 2.37\%$ | 25.49% $\pm 1.38\%$ | 59.76% $\pm 0.86\%$ | 53.18% |
| Qwen2.5 7B | 88.02% $\pm 2.25\%$ | 46.50% $\pm 0.71\%$ | 87.80% $\pm 0.12\%$ | 16.94% $\pm 2.49\%$ | 64.90% $\pm 1.22\%$ | 71.78% $\pm 4.22\%$ | 62.66% |
| Qwen3 8B | **88.21%** $\pm 2.09\%$ | 46.20% $\pm 0.28\%$ | 88.40% $\pm 0.12\%$ | 23.13% $\pm 8.39\%$ | 69.81% $\pm 1.41\%$ | 76.24% $\pm 2.09\%$ | 65.33% |
| *Diffusion Models* | | | | | | | |
| LLaDA 8B | 84.50% $\pm 1.73\%$ | 36.70% $\pm 2.12\%$ | 84.09% $\pm 2.11\%$ | 21.39% $\pm 1.20\%$ | 32.16% $\pm 3.49\%$ | 35.40% $\pm 0.00\%$ | 49.04% |
| Dream 7B | 84.31% $\pm 0.22\%$ | 42.90% $\pm 0.42\%$ | 84.95% $\pm 0.05\%$ | 25.42% $\pm 0.04\%$ | 53.23% $\pm 4.00\%$ | 57.24% $\pm 5.06\%$ | 58.01% |
| ReFusion 7B | 85.64% $\pm 0.59\%$ | 42.90% $\pm 0.71\%$ | 87.98% $\pm 0.49\%$ | 30.45% $\pm 0.09\%$ | 54.12% $\pm 6.64\%$ | 62.05% $\pm 1.10\%$ | 60.52% |
| + Sequential | 77.56% $\pm 2.03\%$ | 38.90% $\pm 0.71\%$ | 87.09% $\pm 0.88\%$ | 33.43% $\pm 0.71\%$ | 50.58% $\pm 0.00\%$ | 62.18% $\pm 2.21\%$ | 58.29% |
| + Random | 66.87% $\pm 1.93\%$ | 29.10% $\pm 0.71\%$ | 80.98% $\pm 0.60\%$ | 27.63% $\pm 3.54\%$ | 21.40% $\pm 1.65\%$ | 32.60% $\pm 1.13\%$ | 43.10% |
| MCDIFFUSE | 87.91% $\pm 1.02\%$ | **47.80%** $\pm 0.85\%$ | **88.68%** $\pm 0.39\%$ | **34.78%** $\pm 0.27\%$ | **73.57%** $\pm 1.61\%$ | **78.37%** $\pm 0.45\%$ | **68.52%** |

*Table 1.* Performance comparison on reasoning and code generation benchmarks (mean and standard deviation). MCDIFFUSE achieves the highest average performance, outperforming both autoregressive and diffusion models. Notably, MCDIFFUSE demonstrates superior performance on MATH500, ARC, GPQA-Diamond, MBPP, and HumanEval, while remaining highly competitive on GSM8K.

# 4. Experimental Setup

**Evaluation Benchmarks and Metrics.** We evaluate on six benchmarks, ranging from mathematical reasoning, code generation, and general knowledge: GSM8K (Cobbe et al., 2021), MATH500 (Hendrycks et al., 2021), MBPP (Austin et al., 2021b), HumanEval (Chen et al., 2021), ARC Challenge (Clark et al., 2018), and GPQA-Diamond (Rein et al., 2023). For all evaluations, we use the Pass@1 metric. To ensure fair comparison, we re-evaluate all baselines using chain-of-thought prompting (Kojima et al. 2022; i.e., let's think step by step), allowing models to generate intermediate reasoning steps before producing final answers. All results averaged over three independent runs and reported with standard errors. (Details appear in Section C.)

**Models.** We use ReFusion (Li et al., 2025) as our base model, as it is, to the best of our knowledge, the only available slot-and-infill diffusion language model. We compare MCDIFFUSE against eight baselines: the vanilla ReFusion model, ReFusion with random slot ordering and sequential slot ordering, LLaDA-8B-Instruct (Nie et al., 2025) and Dream 7B (Ye et al., 2025) as diffusion-based baselines, with Qwen2.5 7B (Team et al., 2024) and Qwen3 8B (Team, 2025) as our autoregressive baseline.

**Hyperparameters.** For the hyperparameters mentioned in Section 3.2, we set $\lambda = 0.3$ to penalise slot orderings that achieve high immediate confidence and force subsequent slots into low-probability regions, guiding the search towards globally coherent generation trajectories. We also use $c = 50$, $N_{sim} = 256$, and $\tau = 0.5$. Detailed explanations of hyperparameter choice and sensitivity analysis are presented in Section 6 and Section C.2.

# 5. Experimental Results

Table 1 reports the performance of MCDIFFUSE across six benchmarks, compared against both masked diffusion models (MDMs) and autoregressive models (ARMs). The results highlight three main phenomena:

**1. MCDIFFUSE outperforms both MDMs and ARMs baselines across five out of six benchmarks.** *Additionally, MCDIFFUSE significantly outperforms all MDMs across all six benchmarks.* For instance, MCDIFFUSE achieves performance increases of 25.98% on HumanEval and 4.00% on MATH500 compared to ReFusion, with more substantial gains on traditional MDMs. (Li et al., 2025).

**2. Coding tasks benefit more from MCTS slot planning.** MCDIFFUSE yields substantially larger gains on code-generation benchmarks, achieving improvements of 16.32% on HumanEval and 19.45% on MBPP. In contrast, improvements on multiple-choice reasoning benchmarks are more modest (ARC Challenge: 0.7%; GPQA: 4.33%). This pattern suggests that tasks involving structured program synthesis benefit more from strategic slot planning, as dependencies between code components—such as variable declarations, function definitions, and control flow—induce ordering constraints that can be effectively explored by MCTS. Further analysis in Section D.2 further confirms this hypothesis.

**3. MCDIFFUSE produces more compact reasoning.** Under identical experimental conditions with chain-of-thought prompting, ARMs often produce lengthy reasoning sequences that may become incoherent or incomplete, particularly when the reasoning chain exceeds the model's effective context length on tasks, like GPQA-Diamond, that require extensive reasoning (see Section E for some qualita-

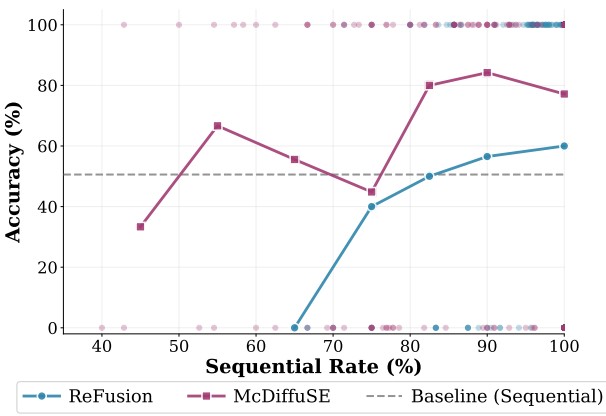

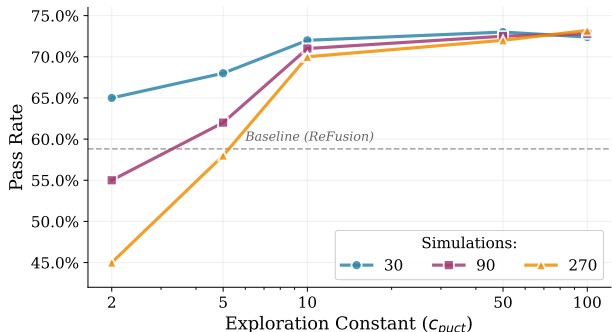

*Figure 3.* Impact of exploration constant ($c$) and simulation budget ($N_{sim}$) on task performance.

*Figure 2.* Relationship between generation sequentiality and accuracy on the MBPP dataset. Each dot represents a sample plotted by its accuracy and sequentiality rate. Darker dots denote higher density, reflecting multiple instances with identical sequentiality. Solid lines denote average accuracy trends computed by binning sequentiality rates for ReFusion and MCDIFFUSE, while the dashed line indicates the overall accuracy of sequential (left-to-right) baseline.

| Config | Mean H (bits) | Std H | Median H | Concentration |
|--------|---------------|-------|----------|---------------|
| *Low Exploration ($c = 2$)* | | | | |
| $N = 30$ | 1.4062 | 0.3785 | 1.4824 | 0.4958 |
| $N = 270$ | 1.1842 | 0.4043 | 1.1492 | 0.6118 |
| *High Exploration ($c = 100$)* | | | | |
| $N = 30$ | 1.4176 | 0.3771 | 1.5058 | 0.4954 |
| $N = 270$ | 1.4340 | 0.3768 | 1.5211 | 0.4879 |

*Table 2.* Impact of simulation count ($N_{sim}$) and exploration constant ($c$) on policy entropy, $H$ and concentration. Under low exploration $c = 2$, entropy decreases with more simulations, while under high exploration $c = 100$, entropy remains stable.

tive examples). In contrast, MCDIFFUSE generates more compact reasoning. For example, on MATH500 MCDIFFUSE produces sequences of average length equal to 152.2 while Qwen2.5 7B has average prediction length equal to 436.0 while getting worse accuracy. The length reduction is consistent across the three benchmarks that require extensive reasoning (i.e., GSM8K, MATH500 and GPQA) and is always statistically significant ($p < 0.001$) as reported in Section D.1. We compare the performance of MCDIFFUSE and autoregressive models at their maximum token lengths in Section D.3.

## 6. Analysis and Discussion

We provide a detailed analysis to better understand when and why MCTS-based slot planning improves generation performance. In particular, we examine how often non-sequential ordering decisions arise and when they matter, how exploration and simulation budget affect accuracy, and what computational trade-offs are introduced by search-based planning. Together, these analyses shed light on the practical behaviour and efficiency of MCDIFFUSE beyond aggregate benchmark results.

**When does non-sequential slot ordering matters?** We observe that the majority of slot-ordering decisions made by MCDIFFUSE follow a sequential, left-to-right pattern, consistent with standard autoregressive decoding (91.1% for coding tasks and 93.8% for mathematical reasoning tasks). As shown in Figure 2, most instances concentrate at high sequentiality rates, indicating that the sequential order serves as an effective default strategy across tasks.

*Despite their relative rarity, non-sequential decisions play a disproportionate role in performance improvements.* Among instances where MCDIFFUSE succeeds while the sequential autoregressive baseline fails (13.2% of the dataset), 60.7% involve at least one non-sequential ordering decision. This effect is also reflected in Figure 2, where higher accuracy is often associated with intermediate sequentiality rates rather than strictly sequential generation. Together, these observations suggest an effective exploration-exploitation trade-off: the search policy largely exploits the reliable sequential prior, while selectively deviating from it to resolve challenging constraints that confound greedy left-to-right decoding. This indicates that even sparse, well-timed departures from the sequential order can yield meaningful accuracy improvements when guided by informed slot planning. Further example and analysis are provided in Figure 4 and Section G.

**How do exploration and simulation budgets affect performance?** We analyse the interaction between the exploration constant ($c$) and the simulation budget ($N_{sim}$) and their impact on the model's performance.

As shown in Figure 3 and Figure 8 in Section I, increasing the exploration budget $c$ consistently improves accuracy across benchmarks, indicating that stronger exploration is

| Model | GSM8K | MATH500 | ARC_C | GPQA-Diamond | MBPP | HumanEval | Average |
|---|---|---|---|---|---|---|---|
| ReFusion | 6.18E+16 | 2.78E+16 | 1.00E+16 | 2.97E+15 | 1.16E+16 | 4.22E+15 | 1.95E+16 |
| MCDIFFUSE | 8.95E+16 | 3.62E+16 | 8.21E+15 | 2.76E+15 | 1.53E+16 | 5.60E+15 | 2.63E+16 |

*Table 3.* Total FLOPs comparison between ReFusion and MCDIFFUSEacross benchmarks.

necessary to overcome the local confidence bias induced by the baseline's predictions.

On the other hand, increasing the simulation budget ($N_{sim}$) does not guarantee an improvement in performance: at low exploration budget ($c = 2.0$) increasing $N_{sim}$ from 30 to 270 simulations leads to a decrease in accuracy on both MBPP and MATH500. This phenomenon is explained by the entropy analysis in Table 2: under low exploration ($c = 2$), mean entropy drops from 1.41 to 1.18 bits as $N_{sim}$ increases, indicating premature convergence towards locally confident but globally myopic orderings. Conversely, with high exploration ($c = 100$), entropy remains stable $\approx 1.43$bits), allowing the search to maintain breadth and discover effective orderings. This suggests that, under insufficient exploration pressure, additional simulations reinforce early, locally confident but globally myopic slot orderings.

Regarding computational overhead, generation time scales approximately linearly with $N_{sim}$ while varying $c$ causes negligible cost as it only affects selection without requiring additional forward passes. We observe an accuracy-efficiency trade-off: although $N_{sim} = 270$ achieves highest accuracy, the improvement over $N_{sim} = 30 < 2\%$ while computational cost increases nine-fold. A high-exploration, low-budget configuration achieves substantial gains such as 13.62% improvement on MBPP while maintaining reasonable inference time. (Detailed analysis and comparison with ReFusion, together with further FLOP analysis, can be found in Section H.)

These results highlight that the primary challenge in non-autoregressive slot planning is not search depth, but sufficient breadth to escape local optima.

**How does MCTS mitigate local optima in slot-based generation?** We give an intuition through the qualitative example in Figure 4. In the Figure, ReFusion selects and fills multiple high-confidence slots simultaneously from the start (slots 1, 2, and 3) in its attempt to maximise throughput. Specifically, it greedily fills slot 1 with an unnecessary import statement (i.e., "Import reverse_"), likely driven by a generic prior that Python scripts often begin with imports. However, because ReFusion cannot backtrack to correct this early commitment, this unnecessary slot forces the subsequent generation into a syntactically broken state (e.g., merging the import line directly into a part of a function definition header). On the other hand, MCDIFFUSE avoids this issue through its lookahead capability. By simu-

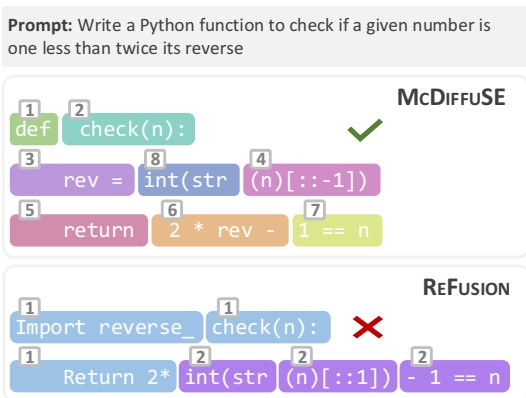

*Figure 4.* Comparison of ReFusion and MCDIFFUSE on a coding prompt from MBPP. Superscripts denote the infilling slot order and colours indicate the specific generation step.

lating the full trajectory before committing, it detects that starting with an import leads to lower long-term coherence for this specific task, and instead correctly prioritises the function definition (by filling in slot 1 with "def" and then slot 2 with "check(n):" to complete the header of the function definition) as the most effective starting point.

**How does MCDIFFUSE compare to ReFusion in computational cost?** As shown in Table 3, MCDIFFUSE incurs moderately higher FLOPs than ReFusion on most benchmarks, with the largest increases on GSM8K, MATH500, MBPP, and HumanEval, while requiring marginally fewer FLOPs on GPQA-Diamond and ARC_C. On average, MCDIFFUSE uses approximately 35% more compute than ReFusion ($1.95E + 16 \rightarrow 2.63E + 16 FLOPs$). Crucially, this additional cost is accompanied by consistent accuracy gains across all evaluated benchmarks, indicating a favourable compute–performance trade-off.

# 7. Related work

**Monte Carlo tree search.** MCTS (Kocsis & Szepesvári, 2006; Coulom, 2006) has proven highly effective for sequential decision-making (Chaslot et al., 2008; Browne et al., 2012; van Krieken et al., 2025), achieving notable success in game-playing agents such as AlphaGo (Granter et al., 2017) and AlphaZero (Silver et al., 2017). Recently, MCTS has been extensively applied to enhance LLM Reasoning, including math (Gao et al., 2024; Xie et al., 2024; Wu et al., 2024), coding (Wang et al., 2025), and others. MCTS has

also been applied to visual diffusion models (Yoon et al., 2025a;b; Ramesh & Mardani, 2025) for improving generation quality. To the best of our knowledge, our work represents the first application of MCTS to MDMs for text generation, specifically addressing slot ordering.

**Masked diffusion models (MDMs).** MDMs, originally proposed by Austin et al. (2021a), have been recently simplified and demonstrated at scale into a promising paradigm for non-autoregressive sequence generation (Shi et al., 2024; Sahoo et al., 2024; Ou et al., 2025), particularly within the coding domain (Gong et al., 2025; Zhao et al., 2025). However, MDMs still struggle to match the performance of autoregressive models (ARMs) on complex reasoning tasks (Zhu et al., 2025). Architectures such as BD3LM (Arriola et al., 2025), Eso-LMs (Sahoo et al., 2025), and ReFusion (Li et al., 2025) have explored various decoding approaches to bridge this gap. Probabilistic methods such as top_k (Nie et al., 2025), low entropy (Ben-Hamu et al., 2025) and probability margins between top candidates (Kim et al., 2025) have been widely used in decoding within text MDMs.

## 8. Conclusion

We introduce McDiffuse, a training-free framework that enhances MDMs through strategic slot selection. By formulating slot ordering as decision-making and using MCTS with designated adaptations – confidence-aware value propagation and adaptive exploration budgets – McDiffuse systematically navigates the combinatorial space of generation orders without additional training. Through experiments across six reasoning benchmarks, McDiffuse demonstrates consistent improvements over existing MDM baselines and competitive performance with ARMs under identical experimental conditions, with particularly strong gains on code generation tasks. Our analysis reveals that (1) while McDiffuse mainly adheres to sequential ordering, strategic deviations to non-sequential generation are essential for maximising performance, and (2) rather than increasing the number of simulations, significant exploration breadth is needed to overcome the model's priors and discover effective slot orderings. Together, these findings establish McDiffuse as an effective approach for enhancing slot selection and infilling within MDMs.

## Impact Statement

This paper presents work whose goal is to apply a controlled "plan-and-infill" search method using MCTS to masked diffusion models without requiring additional training, with applications in mathematical reasoning, coding, and general scientific reasoning tasks within language models. This could improve how diffusion models process generations by exploring multiple search paths with lookahead, reducing

propagation of unmodelled dependencies across iterations that undermine global coherence. Our analysis on models requiring a combination of sequential and non-sequential generation strategies for maximised performance could be useful for future work on improving LLM reasoning, merging the realm between diffusion models and autoregressive models. While currently slots in text diffusion may not be the best application on the speed-accuracy frontier, we observe it is most impactful in settings where reasoning trajectories allow revision or alternative orderings, enabling adaptive compute allocation that mitigates early mistakes and improves long-term decision quality beyond strict left-to-right generation. Lastly, our results on large exploration constants being necessary to overcome the model's confidence priors could bring a societal impact towards the potential of structured "plan-and-infill" methods, or any diffusion-guided generation paradigms. While our approach is built around slot-based infilling, and our evaluations therefore focus on ReFusion, we agree that extending MCTS slot planning to other DLM/MDM architectures remains an important direction. There are potential societal consequences of our work, none of which we feel must be specifically highlighted here.

## Acknowledgements

This work was supported by the AI for Math Fund, which is managed by Renaissance Philanthropy in partnership with founding donor XTX Markets. We thank the anonymous reviewers and area chairs for their helpful comments and feedback. We thank Cyrus Kwan and Ana Ezquerro for the constructive feedbacks on the paper. We are also grateful for the compute resources provided to us by the University of Edinburgh (Edinburgh International Data Facility), and UKRI (Isambard AI service, University of Bristol). Yu Zhao is partly supported by the UKRI Centre for Doctoral Training in Natural Language Processing, funded by UK Research and Innovation (grant EP/S022481/1) and the University of Edinburgh, School of Informatics.

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

# A. Details of the MCDIFFUSE Algorithm

This section details the constituent phases of the MCDIFFUSE algorithm. While the main text above outlines the stochastic confidence rollout used for value estimation (Algorithm 1), here we provide the complete pseudo-code for selecting the next slot.

In particular, Algorithm 2 outlines the overall slot selection framework. The specific subroutines are detailed in Algorithm 3 (SELECT), Algorithm 4 (EXPAND), and Algorithm 5 (BACKPROPAGATE). Note that, in Algorithm 2, the ROLLOUT function call refers directly to Algorithm 1. Additionally, the SELECTCHILD function applies the same PUCT criterion as Algorithm 3 (SELECT), but performs only a single selection step from the current node.

---

**Algorithm 2** MCTS-based Slot Selection for MCDIFFUSE

---

1: **Input:** Current state $s_t = (\sigma_t, \mathbf{x}_t)$, slot-level confidence scores $\{\mathcal{R}(s_t, a)\}_{a \in \mathcal{A}_t}$
2: **Parameters:** Number of simulations $N_{\text{sim}}$, exploration constant $c$, temperature $\tau$, mixing coefficient $\lambda$
3: **Output:** Next action $a^* \in \mathcal{A}_t$ (slot index to fill next)
4:   *// Initialise root node and global statistics*
5:   $s_{\text{root}} \leftarrow s_t$
6:   $N_{s_t} \leftarrow 0$ *// Total visits to root state*
7:   **for all** $a \in \mathcal{A}_t$: $N_{s_t}^a \leftarrow 0, W(s_t, a) \leftarrow 0$
8:   *// Normalise confidence scores to obtain priors*
9:   $\{P(a \mid s_t)\}_{a \in \mathcal{A}_t} \leftarrow \text{Normalise}(\mathcal{R}(s_t, a)_{a \in \mathcal{A}_t})$
10: **for** $n \leftarrow 1$ **to** $N_{\text{sim}}$ **do**
11:     *// 1. Selection: traverse tree using PUCT to reach a leaf*
12:     $s_{\text{leaf}} \leftarrow \text{SELECT}(s_{\text{root}}, c)$
13:     *// 2. Expansion: if leaf is non-terminal and unexpanded, expand it*
14:     **if** $s_{\text{leaf}}$ is not terminal **and** $s_{\text{leaf}}$ has no children **then**
15:       $\text{EXPAND}(s_{\text{leaf}}, \{\mathcal{R}(s_{\text{leaf}}, a)\})$ *// Creates children, initialises stats*
16:       $s_{\text{eval}} \leftarrow \text{SELECTCHILD}(s_{\text{leaf}}, c)$ *// Pick one child*
17:     **else**
18:       $s_{\text{eval}} \leftarrow s_{\text{leaf}}$
19:     **end if**
20:     *// 3. Simulation: mix immediate confidence with rollout value*
21:     $V \leftarrow \lambda \cdot \mathcal{R}(s_{\text{leaf}}, s_{\text{eval}}.\text{action}) + (1 - \lambda) \cdot \text{ROLLOUT}(s_{\text{eval}}, \tau, \{\mathcal{R}(s_{\text{eval}}, a)\})$     *// $s_{eval}$.action is the action that led to $s_{eval}$*
22:     *// 4. Backpropagation: update statistics to root*
23:     $\text{BACKPROPAGATE}(s_{\text{eval}}, V, s_{\text{root}})$
24: **end for**
25: *// Select action with most visits*
26: $a^* \leftarrow \arg\max_{a \in \mathcal{A}_t} N_{s_t}^a$
27: **return** $a^*$

---

**Algorithm 3** SELECT

---

1: **Input:** Current state $s$, exploration constant $c$
2: **Output:** Selected leaf node
3: **while** $s$ has children **do**
4:     *// Calculate PUCT for all valid actions a from state s*
5:     $a^* \leftarrow \arg\max_{a \in \mathcal{A}(s)} \left( Q(s, a) + c \cdot P(a \mid s) \frac{\sqrt{N_s}}{1 + N_s^a} \right)$
6:     $s \leftarrow \text{child}(s, a^*)$    *// Traverse to the child node corresponding to action $a^*$*
7: **end while**
8: **return** $s$

---

# B. Stochastic Confidence Rollout Example

As described in Section 3, we use stochastic confidence rollouts to estimate the long-term value of partial generation states during MCTS simulation. In particular, Algorithm 1 performs the temperature-scaled stochastic sampling over remaining unfilled slots, where slots with higher model confidence receive proportionally higher selection probability but are not deterministically chosen. This stochasticity enables exploration of diverse completion paths, capturing the model's uncertainty about optimal slot orderings while maintaining bias towards high-confidence regions of the generation space. This section provides the detailed algorithm and a concrete numerical example to illustrate the rollout mechanism.

---

**Algorithm 4** EXPAND

---

1: **Input:** State $s$, slot-level confidence scores $\{\mathcal{R}(s, a)\}$
2: *// Identify valid actions (indices of currently unfilled slots)*
3: $\mathcal{A}_s \leftarrow \{$indices of unfilled slots in $s\}$
4: $R_{\text{total}} \leftarrow \sum_{a \in \mathcal{A}_s} \mathcal{R}(s, a)$
5: *// Create a child node for each $a$ and initialise its statistics*
6: **for** each action $a \in \mathcal{A}_s$ **do**
7: $\quad N_s^a \leftarrow 0, \quad W(s, a) \leftarrow 0$
8: $\quad P(a \mid s) \leftarrow \mathcal{R}(s, a)/R_{\text{total}}$ $\quad$ *// Normalise to get prior probability*
9: $\quad$ Create child node $s'$ for action $a$ and link as child$(s, a)$
10: **end for**

---

---

**Algorithm 5** BACKPROPAGATE

---

1: **Input:** State $s$, values $V$ estimated during rollout, root $s_{\text{root}}$
2: **while** $s \neq s_{\text{root}}$ **do**
3: $\quad p \leftarrow$ parent of $s$ $\quad$ *// Get parent node of s*
4: $\quad a \leftarrow$ action from $p$ to $s$ $\quad$ *// Get action that led from p to s*
5: $\quad$ *// Update statistics for the edge $(p, a)$*
6: $\quad N_p^a \leftarrow N_p^a + 1$
7: $\quad W(p, a) \leftarrow W(p, a) + V(p, a)$
8: $\quad Q(p, a) \leftarrow W(p, a)/N_p^a$
9: $\quad$ *// Update total visits for parent state*
10: $\quad N_p \leftarrow N_p + 1$
11: $\quad s \leftarrow p$ $\quad$ *// Move up the tree*
12: **end while**

---

Consider a state where slots 0 and 1 are filled, and we must evaluate the path forward for slots $\{2, 3, 4\}$ with confidences $R(s, 2) = 0.45$, $R(s, 3) = 0.82$, $R(s, 4) = 0.63$ and temperature $\tau = 0.5$.

**Iteration 1:** The rollout computes the sampling distribution:

$$p_2 = \frac{\exp(0.45/0.5)}{\sum} = \frac{2.460}{11.140} = 0.221,$$

$$p_3 = \frac{\exp(0.82/0.5)}{\sum} = \frac{5.155}{11.140} = 0.463,$$

$$p_4 = \frac{\exp(0.63/0.5)}{\sum} = \frac{3.525}{11.140} = 0.316.$$

The algorithm samples $a^* = 3$ (higher confidence slot has higher probability but selection is stochastic). Accumulated reward: $V_{\text{total}} = 0.82$, $n = 1$.

**Iteration 2:** With remaining slots $\{2, 4\}$:

$$p_2 = 0.411, \quad p_4 = 0.589.$$

Sample $a^* = 4$. Update: $V_{\text{total}} = 1.45$, $n = 2$.

**Iteration 3:** Only slot 2 remains, select deterministically. Final: $V_{\text{total}} = 1.90$, $n = 3$.

**Return:** $V(s) = 1.90/3 = 0.633$, representing the expected quality of completing generation from this state.

This stochastic sampling enables the rollout to explore different completion paths, capturing the model's uncertainty about optimal slot ordering.

## C. Experimental Setup

### C.1. Additional Implementation Details

To ensure fair comparison across all methods, we maintain consistent hyperparameters throughout our experiments. All models use an evaluation length of 512 tokens following the configuration from (Li et al., 2025), with increasing token to

1024 for MBPP due to task complexity. We employ greedy decoding with temperature 0.0 across all settings, with zero-shot learning. For ReFusion (Li et al., 2025), we adopt the default configuration: slot size 8, 2 serial blocks, slot threshold 0.9, and token threshold 0.9. For MCDIFFUSE, we configure 30 MCTS simulations, slot size 4, 32 serial blocks, slot threshold 0.5, token threshold 0.6, and exploration constant $c = 10.0$. Autoregressive baselines employ greedy decoding with temperature 0.0 and top_p=1.0. All evaluations are conducted on 4 NVIDIA H100 GPUs. We average all across 3 seeds for consistency, using Seed $= 21, 42, 84$.

### C.2. Additional Hyperparameter Tuning

We further perform an additional hyperparameter tuning on the temperature, $\tau$, and the mixing coefficient $\lambda$. However, due to computational cost, we are unable to find the optimal grid search using all four hyperparameters stated in Section 6. We take a hyperparameter tuning of $c_{puct}$ of 50.0 with simulation of 256 to find the optimal budget. We perform the hyperparameter tuning on MATH-500 dataset.

*Table 4.* Grid Search Results for MATH500: Accuracy (%) by $\tau$ and $\lambda$

| $\tau \setminus \lambda$ | 0.0 | 0.3 | 0.5 | 0.7 | 0.9 | 1.0 |
|---|---|---|---|---|---|---|
| 0.1 | 43.20 | 44.40 | 43.80 | 44.00 | 43.40 | 43.00 |
| 0.3 | 44.80 | 47.00 | 46.20 | 46.40 | 45.20 | 44.60 |
| 0.5 | 45.60 | **48.20** | 47.20 | 47.40 | 45.80 | 44.80 |
| 0.7 | 45.00 | 47.20 | 46.40 | 46.60 | 45.40 | 44.20 |
| 1.0 | 43.60 | 44.80 | 44.20 | 44.20 | 43.60 | 43.00 |

Based on the table, we can observe that the best hyperparameter tuning lies on $\tau = 0.5$, and $\lambda = 0.3$. This is because strikes a critical balance in the exploration-exploitation trade-off: it introduces sufficient stochasticity to prevent the search from converging prematurely on local optima (a failure mode observed at lower temperatures), while maintaining enough coherence to avoid the semantic degeneration typical of higher entropy distributions.

Furthermore, $\lambda = 0.3$ indicates that the value estimation benefits significantly from prioritising long-term rollout feedback over immediate confidence. In our formulation, a lower $\lambda$ shifts the weight towards the Monte Carlo rollout return rather than the immediate policy prior. This suggests that for complex mathematical reasoning, the verifiability of a full solution path (the rollout) is a more reliable signal than the model's local token probabilities, though the immediate confidence remains necessary as a regularising prior to guide the initial search direction.

## D. Experimental Results and Tests

### D.1. Analysis Between Token Length and Accuracy

*Table 5.* Token efficiency and accuracy comparison between MCDIFFUSE and Qwen2.5-7B across benchmark datasets. All reductions are statistically significant ($p < 0.001$).

| | **Token Count** | | | **Accuracy (%)** | | |
|---|---|---|---|---|---|---|
| **Dataset** | MCDIFFUSE | Qwen2.5 | Reduction | MCDIFFUSE | Qwen2.5 | Improvement |
| MBPP | 44.0 | 368.2 | $-88.05\%$ | 72.43 | 64.04 | $+8.39$ |
| ARC | 91.3 | 332.8 | $-72.55\%$ | 88.40 | 87.88 | $+0.52$ |
| MATH500 | 152.2 | 436.0 | $-65.08\%$ | 48.40 | 46.00 | $+2.40$ |
| GSM8K | 184.0 | 276.2 | $-33.39\%$ | 88.63 | 86.43 | $+2.20$ |
| GPQA | 34.6 | 534.3 | $-93.53\%$ | 34.78% | 16.94% | $+17.84$ |
| HumanEval | 63.6 | 256.3 | $-75.19\%$ | 78.37% | 71.78% | $+6.59$ |

Table 5 demonstrates that MCDIFFUSE achieves a significant improvements in both token efficiency and accuracy compared to the autoregressive baseline. Across all benchmarks, MCDIFFUSE reduces token consumption by an average of 64.77% ($p < 0.001$) while simultaneously improving accuracy by 3.38% percentage points. The efficiency gains are particularly great on code generation tasks, with MBPP exhibiting an 88.05% token reduction alongside an 8.39% improvement. This

pattern suggests that MCDIFFUSE is especially effective at identifying compact reasoning paths for structured generation tasks. Even on datasets where accuracy improvements are modest (e.g., ARC with $+0.52\%$), the token reduction remains substantial at $72.55\%$, indicating that MCDIFFUSE consistently produces more concise solutions without sacrificing correctness. The inverse relationship between token reduction and task complexity is evident: while MBPP and ARC show the largest reductions ($88.05\%$ and $72.55\%$ respectively), GSM8K exhibits a more moderate $33.39\%$ reduction, likely due to the sequential nature of multi-step arithmetic reasoning. Nevertheless, the consistent improvements across all benchmarks validate that MCTS-guided slot selection enables more efficient exploration of the solution space compared to purely confidence-based ordering. Figure 5 provides a clear visualisation of the token reduction.

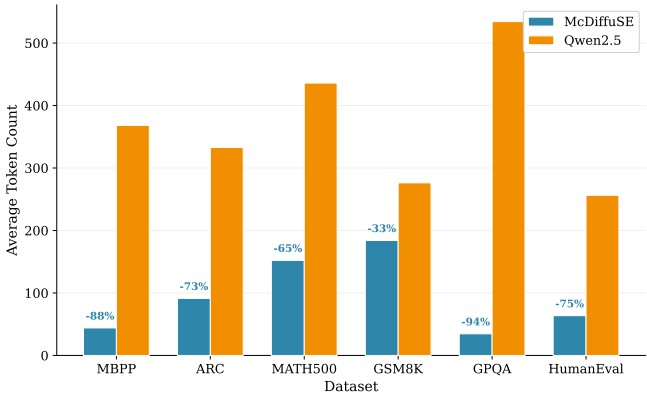

*Figure 5.* Comparison of token reduction across models. We observe that MCDIFFUSE significantly reduces tokens while generating responses, demonstrating the compactness and coherence of using MCDIFFUSE.

### D.2. Task-Specific Improvements

To investigate whether MCTS-based slot planning benefits different task types differentially, we analyse the absolute accuracy improvements from ReFusion to MCDIFFUSE across our evaluation benchmarks. Coding tasks (MBPP, HumanEval) demonstrate substantially larger improvements compared to reasoning and multiple-choice tasks. As shown in Table 6, this difference is statistically significant ($t = 2.97$, $p = 0.041$, Cohen's $d = 3.21$), with coding tasks exhibiting $3.13\times$ higher improvements. *We observe similar behaviour with Qwen3, given its similar architecture to Qwen2.5.* This suggests that MCTS-based planning particularly benefits domains with strict structural dependencies, such as variable declarations, function definitions, and control flow sequences, where slot ordering critically affects semantic validity.

*Table 6.* Statistical comparison of MCTS improvements between coding and non-coding tasks.

| Metric | Values |
|---|---|
| Difference | $7.43\%$ ($3.13\times$) |
| $t$-statistic | 2.97 |
| $p$-value | 0.041 |
| Cohen's $d$ | 3.21 |
| Significance | $p < 0.05$ ✓ |

### D.3. Impact of Extended Token Length on Autoregressive Models

We compare the total completion token lengths between autoregressive models and MCDIFFUSE. Specifically, we evaluate Qwen2.5 7B, Qwen3 8B, and MCDIFFUSE by extending their maximum token lengths to match their respective baselines: 32,768 tokens for both Qwen2.5 and Qwen3, and 1,024 tokens for ReFusion. Due to the nature of ReFusion and other existing MDMs (Nie et al., 2025; Zhao et al., 2025; Zhu et al., 2025; Li et al., 2025; Gong et al., 2025), the model's reasoning capabilities could not be extended for a fair comparison; however, we acknowledge that investigating token length extensions for diffusion-based models represents a promising direction for enhancing LLM reasoning capabilities. We restrict our analysis to MATH500, GSM8K, and GPQA-Diamond, as these are the only benchmarks mentioned in Section 4 with incomplete baseline results.

*Table 7.* Comparison of ReFusion-MCTS, Qwen2.5, and Qwen3 benchmarks.

| Model | Context | GSM8K | MATH500 | GPQA |
|---|---|---|---|---|
| MCDIFFUSE | 1024 | **90.63%** | 54.60% | 37.19% |
| Llama3.1 7B | 8096 | 84.91% | 39.40% | 37.71% |
| Qwen2.5 7B | 32768 | 89.99% | 63.40% | 35.94% |
| Qwen3 8B | 32768 | 88.17% | **68.00%** | **41.96%** |

Table 7 demonstrates that MCDIFFUSE achieves competitive performance while using significantly fewer tokens (1024 vs 32,768). Specifically, MCDIFFUSE marginally outperforms both Qwen2.5 7B and Qwen3 8B on GSM8K. On GPQA, MCDIFFUSE surpasses Qwen2.5 but underperforms relative to Qwen3 8B. However, MCDIFFUSE exhibits weaker performance on MATH500, a benchmark requiring complex mathematical reasoning. Extending ReFusion to 4096 tokens leads to the same performance. In this case, we attribute this limitation to the inherent constraint of ReFusion and existing diffusion models, which cannot generate sequences beyond their predetermined token limit. We hypothesise that extending the maximum token length of diffusion-based models could substantially improve their reasoning capabilities.

### D.4. Clarification on the Difference Between ReFusion Baselines

| Model | GSM8K | MATH500 | ARC | GPQA-Diamond | MBPP | HumanEval | Average |
|---|---|---|---|---|---|---|---|
| ReFusion | 84.91 | 54.00 | 89.76 | 32.32 | 68.20 | 78.66 | 67.98 |
| DiffuSearch | 88.10 | 62.20 | 91.98 | 36.87 | 78.40 | 89.63 | 74.53 |

*Table 8.* Performance comparison across reasoning, coding, and scientific benchmarks.

In this section we provide a clarification between the differences between our reported baselines in section 5. Specifically, ReFusion the original ReFusion (Li et al., 2025) applies Harness-Evaluation. which uses task-specific (i) prompts, (ii) answer extraction, and (iii) parsing logic. In contrast, our initial experiments used the standard CoT prompt [2] (i.e. "let's think step by step"). This was done intentionally to observe the reasoning traces and study the effect of introducing MCTS under a consistent prompting setup. We further re-evaluated all baselines using LM-Eval-Harness (Biderman et al., 2024) with matched decoding and configurations, and based on Table 8, the improvements remain consistent.

Notably, coding benchmarks (MBPP, HumanEval) show the most substantial gains ($> 10\%$), while multiple-choice benchmarks (ARC-Challenge, GPQA) exhibit marginal improvements. MATH500 also demonstrates a meaningful gain of $+8.2\%$, suggesting that MCTS is particularly effective on tasks requiring extended, multi-step reasoning.

### D.5. Additional Experimental Results and Baselines

| Model | GSM8K | MATH500 | ARC_C | GPQA-Diamond | MBPP | HumanEval | Average |
|---|---|---|---|---|---|---|---|
| ReFusion | 84.91 | 54.00 | 89.76 | 32.32 | 68.20 | 78.66 | 67.98 |
| Best-of-N | 79.76 | 45.60 | 88.65 | 30.81 | 57.20 | 68.90 | 61.82 |
| Beam Search | 81.65 | 49.40 | 89.42 | 31.82 | 59.40 | 71.34 | 63.84 |
| DiffuSearch | 88.10 | 62.20 | 91.98 | 36.87 | 78.40 | 89.63 | 74.53 |

*Table 9.* Accuracy comparison between ReFusion, compute-matched baselines, and DiffuSearch across benchmarks.

We further re-evaluated all baselines using LM-Eval-Harness with matched decoding and configurations, and the improvements remain consistent. Additionally, we include two compute-matched baselines: (1) best-of-N sampling with reranking using the same confidence score, and (2) beam search over slot order decisions. Based on the results on Section D.5, we can conclude that naively scaling inference compute does not guarantee improvement and can even degrade performance relative to baseline ReFusion. This confirms that the gains are not simply a result of increased sampling budget. DiffuSearch, in contrast, consistently improves accuracy across all benchmarks by strategically allocating compute through tree-structured exploration.

# E. Example of ReFusion Generation Length Correspond To Autoregressive

### E.1. Example 1

---

**McDiffuSE Generation** *(Correct, 278 tokens)*

**Question:** A merchant wants to make a choice of purchase between 2 purchase plans: jewelry worth $5,000 or electronic gadgets worth $8,000. His financial advisor speculates that the jewelry market will go up 2.5% while the electronic gadgets market will rise 1.2% within the same month. If the merchant is looking to maximise profit at the end of this month by making a choice, how much profit would this be?
**Ground Truth:** 125

**Generated Solution:**
Step 1: Calculate the future value of the jewelry purchase.
Future value = Current value + (Current value × Percentage increase)
Future value = $5,000 + ($5,000 × 0.025) = $5,000 + $125 = $5,125
Step 2: Calculate the future value of the electronic gadgets purchase.
Future value = $8,000 + ($8,000 × 0.012) = $8,000 + $96 = $8,096
Step 3: Calculate the profit for each purchase plan.
For the jewelry purchase: Profit = $5,125 - $5,000 = $125
For the electronic gadgets purchase: Profit = $8,096 - $8,000 = $96
Step 4: Compare the profits and choose the purchase plan with the highest profit.
The jewelry purchase has a higher profit of $125 compared to the electronic gadgets purchase with a profit of $96.
**The final answer is: $125**
**Answer:** 125

---

**Qwen2.5-7B Generation** *TRUNCATED (512 tokens)*

**Generated Solution (Truncated):**
Step 1: Calculate the value of the jewelry after the market increase

- Initial value of jewelry: $5,000

- Speculated market increase: 2.5%

First, convert the percentage increase to a decimal: $2.5\% = \frac{2.5}{100} = 0.025$
Next, calculate the increase in value: Increase in value $= 5000 \times 0.025 = 125$
Finally, add the increase to the initial value: Final value of jewelry $= 5000 + 125 = 5125$
Step 2: Calculate the value of the electronic gadgets after the market increase

- Initial value of electronic gadgets: $8,000

- Speculated market increase: 1.2%

First, convert the percentage increase to a decimal: $1.2\% = \frac{1.2}{100} = 0.012$
Next, calculate the increase in value: Increase in value $= 8000 \times 0.012 = 96$
Finally, add the increase to the initial value: Final value of electronic gadgets $= 8000 + 96 = 8096$
Step 3: Determine the profit for each purchase
Profit from jewelry: Profit $= 5125 - 5000 = 125$
Profit from electronic gadgets: Profit $= 8096 - 8000 = 96$
Step 4: Compare the profits
Since $125 > 96$, the merchant should choose to purchase the jewelry to **[TRUNCATED]**

---

## E.2. Example 2

---

**McDiffuSE Generation** *(Correct, 319 tokens)*

**Question:** Which of the following actions will increase the gravitational force between two masses, $m_1$ and $m_2$, by the greatest amount?
**Ground Truth:** C

**Generated Solution:**
The gravitational force between two masses is given by the formula $F = G \cdot \frac{m_1 \cdot m_2}{r^2}$, where $G$ is the gravitational constant, $m_1$ and $m_2$ are the masses, and $r$ is the distance between the masses. To increase the gravitational force, we need to increase the numerator $(m_1 \cdot m_2)$ and decrease the denominator $(r^2)$.
**Option (A):** Halving the mass of $m_1$ and halving the distance between $m_1$ and $m_2$ will decrease the numerator by half and decrease the denominator by a quarter, resulting in a net decrease in the gravitational force.
**Option (B):** Halving the mass of $m_1$ and doubling the distance between $m_1$ and $m_2$ will decrease the numerator by half and increase the denominator by a quarter, resulting in a net decrease in the gravitational force.
**Option (C):** Doubling the mass of $m_1$ and halving the distance between $m_1$ and $m_2$ will increase the numerator by a factor of 2 and decrease the denominator by a quarter, resulting in a net increase in the gravitational force.
**Option (D):** Doubling the mass of $m_1$ and doubling the distance between $m_1$ and $m_2$ will increase the numerator by a factor of 2 and increase the denominator by a quarter, resulting in a net increase in the gravitational force.
Comparing the options, we can see that option (C) will result in the greatest increase in the gravitational force, as it increases the numerator by a factor of 2 and decreases the denominator by a quarter.
**The final answer is: C**
**Answer: C**

---

**Qwen2.5-7B Generation** *TRUNCATED (512 tokens)*

**Generated Solution (Truncated):**
To determine which action will increase the gravitational force between two masses, $m_1$ and $m_2$, by the greatest amount, we need to consider the formula for gravitational force:

$$F = G\frac{m_1 m_2}{r^2}$$

where $F$ is the gravitational force, $G$ is the gravitational constant, $m_1$ and $m_2$ are the masses, and $r$ is the distance between the centres of the two masses.
Let's analyse each option step by step:
**Option (A): Halve the mass of $m_1$ and halve the distance between $m_1$ and $m_2$.**
New force:
$$F' = G\frac{\frac{m_1}{2} \cdot m_2}{\left(\frac{r}{2}\right)^2} = G\frac{\frac{m_1 m_2}{2}}{\frac{r^2}{4}} = G\frac{m_1 m_2}{2} \cdot \frac{4}{r^2} = 2 \cdot G\frac{m_1 m_2}{r^2} = 2F$$

The force is doubled.
**Option (B): Halve the mass of $m_1$ and double the distance between $m_1$ and $m_2$.**
New force:
$$F' = G\frac{\frac{m_1}{2} \cdot m_2}{(2r)^2} = G\frac{\frac{m_1 m_2}{2}}{4r^2} = \frac{1}{8} \cdot G\frac{m_1 m_2}{r^2} = \frac{1}{8}F$$

The force is reduced to one-eighth.
**Option (C): Double the mass of $m_1$ and halve the distance between $m_1$ and $m_2$.**
New force:
$$F' = G\frac{2m_1 \cdot m_2}{\left(\frac{r}{2}\right)^2} = G\frac{2m_1 m_2}{\frac{r^2}{4}} = 8 \cdot G\frac{m_1 m_2}{\textbf{[TRUNCATED]}}$$

---

## E.3. Example 3

---

**McDiffuSE Generation** *(Correct, 403 tokens)*

**Question:** Find all values of $x$ that satisfy the equation $x = \sqrt{11 - 2x} + 4$.
**Ground Truth:** $x = 5$

**Generated Solution:**
To solve the equation $x = \sqrt{11 - 2x} + 4$, we will first isolate the square root term and then square both sides of the equation to eliminate the square root.
**Step 1: Isolate the square root term**
$$x = \sqrt{11 - 2x} + 4 \Rightarrow x - 4 = \sqrt{11 - 2x}$$

**Step 2: Square both sides of the equation**
$$(x - 4)^2 = \left(\sqrt{11 - 2x}\right)^2 \Rightarrow x^2 - 8x + 16 = 11 - 2x$$

**Step 3: Rearrange the equation to form a quadratic equation**
$$x^2 - 8x + 16 - 11 + 2x = 0 \Rightarrow x^2 - 6x + 5 = 0$$

**Step 4: Solve the quadratic equation**
$$x^2 - 6x + 5 = 0 \Rightarrow (x - 1)(x - 5) = 0 \Rightarrow x = 1 \text{ or } x = 5$$

**Step 5: Check the solutions in the original equation**
$$x = 1: \quad 1 = \sqrt{11 - 2(1)} + 4 \Rightarrow 1 = \sqrt{9} + 4 \Rightarrow 1 = 3 + 4 \Rightarrow 1 = 7 \text{ (False)}$$
$$x = 5: \quad 5 = \sqrt{11 - 2(5)} + 4 \Rightarrow 5 = \sqrt{1} + 4 \Rightarrow 5 = 1 + 4 \Rightarrow 5 = 5 \text{ (True)}$$

**The final answer is:** $\boxed{5}$
**Answer:** 5

---

**Qwen2.5-7B Generation** *TRUNCATED (512 tokens)*

**Generated Solution (Truncated):**
To solve the equation $x = \sqrt{11 - 2x} + 4$, we will follow these steps:
**1. Isolate the square root term:**
$$x - 4 = \sqrt{11 - 2x}$$

**2. Square both sides to eliminate the square root:**
$$(x - 4)^2 = (\sqrt{11 - 2x})^2$$

Simplifying both sides:
$$(x - 4)^2 = 11 - 2x$$

**3. Expand the left-hand side:**
$$x^2 - 8x + 16 = 11 - 2x$$

**4. Move all terms to one side to form a quadratic equation:**
$$x^2 - 8x + 16 - 11 + 2x = 0$$

Simplifying:
$$x^2 - 6x + 5 = 0$$

**5. Factor the quadratic equation:**
$$(x - 1)(x - 5) = 0$$

**6. Solve for $x$:**
$$x - 1 = 0 \quad \text{or} \quad x - 5 = 0$$
$$x = 1 \quad \text{or} \quad x = 5$$

**7. Verify the solutions by substituting them back into the original equation:**
For $x = 1$:
$$1 = \sqrt{11 - 2(1)} + 4 \Rightarrow 1 = \sqrt{9} + 4 \Rightarrow 1 = 3 + 4 \Rightarrow 1 \neq 7$$

So, $x = 1$ is not a solution.
For $x = 5$:
$$5 = \sqrt{11 - 2(5)} + 4 \Rightarrow 5 = \sqrt{1} + 4\textbf{\textcolor{red}{[TRUNCATED]}}$$

---

# F. Complete MCTS Example Walkthrough

### F.1. Problem Setup

We demonstrate the MCTS algorithm on a MATH500 problem with 4 remaining slots to fill. The model provides slot confidences: $R(s, 0) = 0.35$, $R(s, 1) = 0.78$, $R(s, 2) = 0.52$, $R(s, 3) = 0.41$.

### F.2. Simulation 1: Initial Exploration

**Selection:** Start at ROOT (leaf node, no children).

**Expansion:** Compute priors by normalising confidences:

$$P(a \mid s) = \frac{R(s, a)}{\sum_{a'} R(s, a')}$$
$$P(a = 0 \mid s) = 0.35/2.06 = 0.170, \quad P(a = 1 \mid s) = 0.379,$$
$$P(a = 2 \mid s) = 0.252, \quad P(a = 3 \mid s) = 0.199$$

Create 4 children. Select child with highest PUCT:

$$\text{PUCT}(a) = Q(s, a) + c \cdot P(a \mid s) \cdot \frac{\sqrt{N(s)}}{1 + N(s, a)}$$
$$\text{PUCT}(\text{slot 1}) = 0 + 1.4 \times 0.379 \times \frac{\sqrt{1}}{1 + 0} = 0.531 \quad \text{(max)}$$

**Simulation:** Perform stochastic rollout from state with slot 1 filled:

- Immediate reward: $r_1 = 0.78$

- Rollout: Sample slots $\{2, 3, 0\}$ stochastically, collect rewards $\{0.52, 0.41, 0.35\}$

- Rollout value: $V_{\text{rollout}} = (0.52 + 0.41 + 0.35)/3 = 0.427$

- Combined: $V = 0.3 \times 0.78 + 0.7 \times 0.427 = 0.533$

**Backpropagation:** Update $\text{Child}_1$ and ROOT:

$$N(\text{Child}_1) = 1, \quad Q(\text{Child}_1) = 0.533$$

### F.3. Simulation 2: Deepening the Tree

**Selection:** Recompute PUCT at ROOT with $N(\text{ROOT}) = 1$:

$$\text{PUCT}(\text{slot 0}) = 0 + 1.4 \times 0.170 \times \frac{\sqrt{2}}{1} = 0.337$$
$$\text{PUCT}(\text{slot 1}) = 0.533 + 1.4 \times 0.379 \times \frac{\sqrt{2}}{2} = 0.909 \quad \text{(max)}$$

Select $\text{Child}_1$ again (highest PUCT despite 1 visit).

**Expansion:** $\text{Child}_1$ is now a leaf with filled slots $\{1\}$. Expand with remaining slots $\{0, 2, 3\}$, creating 3 grandchildren.

**Simulation & Backpropagation:** Select $\text{Child}_{1,2}$ (slot 2 after slot 1), perform rollout, get reward $0.422$. Backpropagate to $\text{Child}_{1,2}$, $\text{Child}_1$, and ROOT.

## F.4. Final Selection

After 4 simulations, the tree has visit counts:

$$\text{Visits} = \{\text{slot } 0 : 1, \ \text{slot } 1 : 2, \ \text{slot } 2 : 1, \ \text{slot } 3 : 0\}$$

**Decision:** Select slot 1 (most visits).

## F.5. Key Observation

Although slot 1 had the highest initial confidence (0.78), MCTS confirmed this choice through *exploration*: rollouts from state $\{1\}$ consistently found high-value completion paths, increasing confidence that slot 1 is the optimal first choice.

# G. Analysis Between Sequence Rate and accuracy

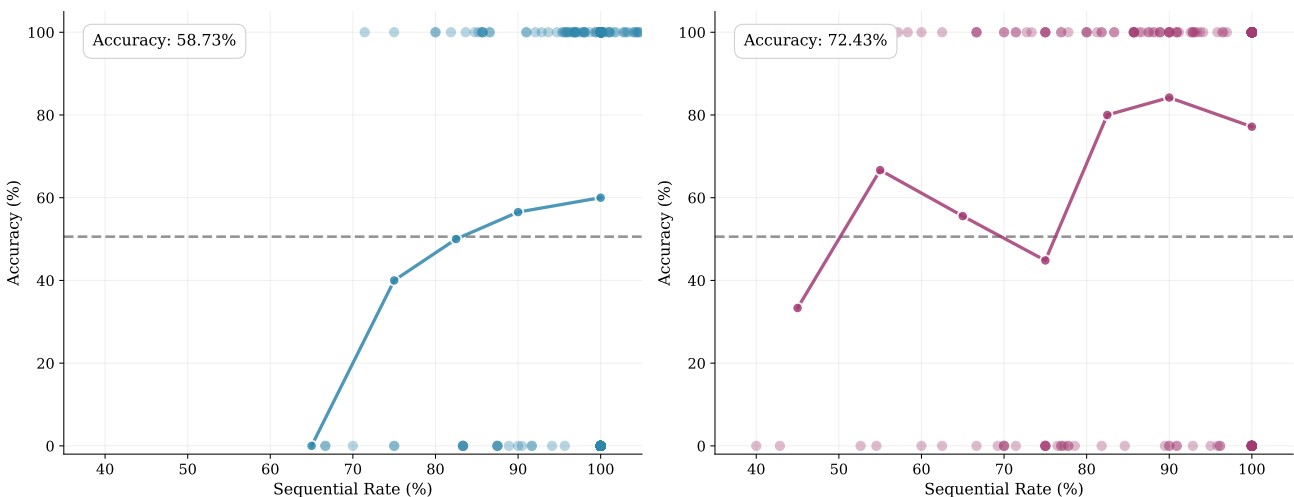

*Figure 6.* Plot with x axis being the sequential rate and y axis being the accuracy. ReFusion exhibits a positive correlation between sequential rate and accuracy, whilst MCDIFFUSE achieves optimal performance at approximately $90\%$ of sequential generation. This demonstrates that while predominantly sequential ordering is beneficial, strategic non-sequential generation leads to improved performance, mirroring human behaviour in coding tasks.

Figure 6 demonstrates a comparative analysis of sequential slot selection behaviour and its relationship to accuracy on the MBPP code generation benchmark for both ReFusion (baseline) and MCDIFFUSE. The baseline ReFusion method (left panel, 58.81% accuracy) shows a weak positive correlation between sequential ordering and task success. In contrast, MCDIFFUSE (right panel, 72.43% accuracy) demonstrates a more strategic approach while achieving substantially higher accuracy. The trend lines reveal that both methods benefit from sequential structure to some degree, consistent with code generation's inherent ordering dependencies (e.g., variable declarations preceding usage); however, MCTS's selective deviation from strict left-to-right generation (8.2% non-sequential exploration) enables it to avoid error propagation and handle complex logical dependencies more effectively. This strategic flexibility, rather than rigid adherence to sequential ordering, accounts for the performance gain, validating the hypothesis that optimal slot planning must adaptively balance sequential structure with non-sequential exploration.

# H. Efficiency Analysis of MCDIFFUSE

In this section, we answer the question: **what is the overhead caused by MCTS-Based Slot Planning?**. We compare the generation time of ReFusion with and without MCTS-based slot planning. As shown in Figure 7, generation time scales approximately linearly with the number of simulations $N_{\text{sim}}$. In contrast, varying the exploration constant $c$ causes negligible overhead, as it only affects the selection criterion without requiring additional forward passes. This observation suggests that as expected computational cost is controlled primarily through the simulation budget.

The data reveals an accuracy-efficiency trade-off. While the highest accuracy is achieved with $N_{\text{sim}} = 270$, the improvement

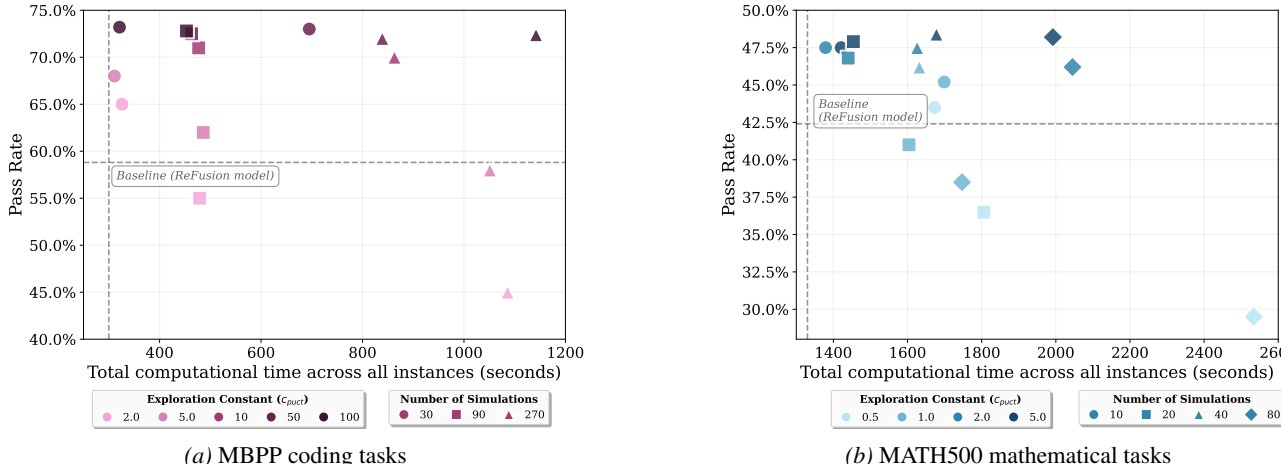

*(a)* MBPP coding tasks      *(b)* MATH500 mathematical tasks

*Figure 7.* Accuracy versus generation time across different hyperparameter settings on coding and mathematical reasoning tasks.

over $N_{\text{sim}} = 30$ is marginal (typically $< 2\%$), whereas the computational cost increases nine-fold. This supports our earlier finding that simulation budgets suffice when exploration is adequate. Although MCTS introduces additional overhead compared to the baseline ReFusion inference, a high-exploration, low-budget configuration (e.g., $c = 100$, $N_{\text{sim}} = 30$) achieves substantial accuracy gains, such as a 13.62% improvement on MBPP, while maintaining reasonable inference time. Hardware details are provided in in Section C.1. In addition to the analysis of coding tasks as show in Figure 7b, we present the analysis using mathematical task MATH500 in Figure 7a. The results show a similar trend: there is a clear linear relationship between the number of simulations $N_{\text{sim}}$ and computational time, while the exploration constant $c$ incurs negligible overhead. While increasing simulations generally improves performance in MCTS, our analysis in Section 5 shows that leveraging broader exploration through CUPT is far more effective, achieving 13.6% performance gains at substantially lower cost. This is because increasing the exploration directs the search toward underexplored but promising reasoning paths without incurring the linear cost of additional rollouts.

## I. More Analysis on the Impact of Exploration and Simulation Budgets

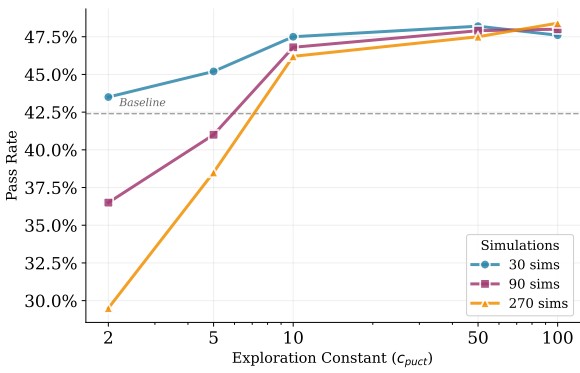

*Figure 8.* Impact of exploration constant ($c_{puct}$) and simulation budget on task performance on MATH500 mathematical reasoning tasks.

In addition to the analysis of coding tasks as shown in Figure 2, we provide the MATH500 mathematical reasoning task in Figure 8. We found a similar trend with coding tasks, where a higher exploration constant obtains higher accuracy.

Based on Figure 8, while we observe that increasing Nsim from 30 to 270 yields diminishing returns (less than 2% improvement at 9× the computational cost). However, while increasing simulations generally improves performance in MCTS, our analysis in Section 5.3 shows that leveraging broader exploration through CUPT is far more effective, achieving 13.6% performance gains at substantially lower cost. This is because increasing the exploration directs the search toward underexplored but promising reasoning paths without incurring the linear cost of additional rollouts.

