# OpenReview forum: "Can I Have Your Order? Monte-Carlo Tree Search for Slot Filling Ordering in Diffusion Language Models"
_ICML.cc/2026/Conference — ICML 2026 regular_

### Official Review · Reviewer_qgAU · 2026-02-22

**Soundness:** 3
**Presentation:** 4
**Significance:** 3
**Originality:** 2
**Overall Recommendation:** 5
**Confidence:** 4

**Summary:**

This paper proposes DiffuSearch, an inference-time framework to perform planning and infilling for masked diffusion language models (DLMs). The key idea is to use Monte Carlo Tree Search (MCTS) to choose which masked slot of tokens to fill next. Empirical results on downstream benchmarks report consistent gains over prior DLM baselines.

**Compliance With Llm Reviewing Policy:**

Affirmed.

**Final Justification:**

I recommend accept. The paper presents a clean and practically useful MCTS-based inference framework for masked diffusion language models, addressing the important issue of infilling/unmasking order. The approach is technically sound, well-motivated, and supported by strong and consistent gains across benchmarks.

My main concerns in the initial review were well addressed by the rebuttal via matched re-evaluation, added compute-matched baselines (best-of-N, beam search), and FLOPs accounting. These reinforce my original positive assessment.

**Key Questions For Authors:**

- What is the sampling strategy for baseline models? Do they generate a single rollout or draw multiple samples? If DiffuSearch uses additional rollouts while baselines do not, could you include roughly compute-matched baselines such as (1) best-of-N sampled slot orderings with reranking by the same confidence score, or (2) simple beam search over slot order decisions or top-k tokens? That will better clarify whether the gains come from the MCTS structure versus the increased sampling search budget.
- Besides wall-clock time, which is an effective metric for throughput comparisons but might depend on the hardware and implementation, are there other impl.-agnostic metrics to compare the efficiency against baselines, such as the number of function evaluations (e.g., number of forward passes) or FLOPs?

**Limitations:**

Not quite. See weaknesses and questions above.

**Strengths And Weaknesses:**

Strengths:
- The paper targets one of the key issues for DLM decoding: the sensitivity to unmasking ordering at inference time, and frames it with a clean and straightforward MCTS formulation.
- Strong empirical results and improved performance-efficiency Pareto frontier: improving task accuracy effectively without incurring significant computational overheads.
- The depth versus breadth discussion and analysis is interesting and provides helpful insights into searching orderings in the context of DLM decoding.

Weaknesses:
- Using confidence as a prior or reward might bias the model toward some unnatural or shortcut solutions, which might correlate with safe / overly short continuations or certain formats.
- The slot / decoding hyperparameters for DiffuSearch versus its baseline ReFusion seem different, making it difficult to isolate how much of the gain comes from the planning versus the decoding or configuration changes.

---

> ### Author Rebuttal · Authors · 2026-03-27
>
> We sincerely thank Reviewer qgAU for recognising DiffuSearch as a clean MCTS formulation that addresses a key issue in DLM decoding, and for acknowledging its strong empirical results without significant computational overhead. We also thank the reviewer for acknowledging the depth versus breadth analysis and its interesting insights into search orderings for DLM decoding.
>
> > **W1:** Using confidence might bias the model toward shortcut solutions.
>
> We thank the reviewer for raising this interesting point! We agree that confidence-based signals could, in principle, bias search toward locally easy-to-predict behaviours such as short continuations or generic formatting. Our choice of confidence is motivated by prior work [1,2], which showed it serves as a strong metric for assigning rewards, and reasoning chain selection. However, our method is designed to mitigate this risk through two mechanisms:
>
> 1. Confidence is not used as a greedy decision rule. The prior only initialises exploration, while action selection is ultimately governed by PUCT and rollout-backed value estimates. This allows the search to override high-confidence but globally poor choices (shown in Figure 1).
> 2. The value function combines immediate confidence with rollout return. We performed a sensitivity analysis in App. C.2, where $ \lambda=0.3$ acts as a sweet spot to mitigate the risks, and places more weight on long-term trajectory quality than on local confidence alone.
>
> Together, these choices ensure confidence guides exploration without dominating final decisions. We will clarify this further in the camera-ready version.
>
> > **W2** Clarification on the difference between hyperparameters for DiffuSearch and baseline ReFusion.
>
> > **Q1** Include reranking (Best-of-N) by confidence score or beam search baseline.
>
> Thank you for bringing this to our attention! We would like to clarify that the discrepancy in baseline results is due to differences in evaluation setup:  ReFusion adopts LM-Eval-Harness [1], which uses task-specific (i) prompts, (ii) answer extraction, and (iii) parsing logic. In contrast, our initial experiments used the standard CoT prompt [2] (i.e. "let's think step by step"). This was done intentionally to observe the generated reasoning traces and study the effect of introducing MCTS under a consistent prompting setup.
>
> We further re-evaluated all baselines using LM-Eval-Harness with matched decoding and configurations, and the improvements remain consistent. Additionally, we include two compute-matched baselines: (1) best-of-N sampling with reranking using the same confidence score, and (2) beam search over slot order decisions:
>
> | Model | GSM8K | GPQA-Diamond | MATH500 | MABPP | HumanEval | ARC_C |
> |---|---:|---:|---:|---:|---:|---:|
> | ReFusion | 84.91% | 32.32% | 54.00% | 68.20% | 78.66% | 89.76% |
> | Best-of-N | 79.76% | 30.81% | 45.60% | 57.20% | 68.90% | 88.65% |
> | Beam Search | 81.65% | 31.82% | 49.40% | 59.40% | 71.34% | 89.42% |
> | DiffuSearch | **88.10%** | **36.87%** | **62.20%** | **78.40%** | **89.63%** | **91.98%** |
>
> Based on the results, we can conclude that naively scaling inference compute does not guarantee improvement and can even degrade performance relative to baseline ReFusion. This confirms that the gains are not simply a result of increased sampling budget. DiffuSearch, in contrast, consistently improves accuracy across all benchmarks by strategically allocating compute through tree-structured exploration. We will include the results in the camera-ready version with further explanation.
>
> > **Q2** Other metrics (i.e., FLOPS) to compare efficiency?
>
> We thank the reviewer for this suggestion. We have evaluated the total FLOPs for both ReFusion and DiffuSearch:
>
> | Model | GSM8K | GPQA-Diamond | MATH500 | MBPP | HumanEval | ARC_C |
> |---|---:|---:|---:|---:|---:|---:|
> | ReFusion | 6.18E+16 | 2.97E+15 | 2.78E+16 | 1.16E+16 | 4.22E+15 | 1.00E+16 |
> | DiffuSearch | 8.95E+16 | 2.76E+15 | 3.62E+16 | 1.53E+16 | 5.60E+15 | 8.21E+15 |
>
> We can see that DiffuSearch often incurs a slight increase in total FLOPs compared to ReFusion, yet this additional compute translates into consistent accuracy gains across all benchmarks. We will include a detailed FLOPs analysis in Appendix H of the camera-ready version. Thank you again for this valuable suggestion!
>
> We hope our responses address your concerns. Please feel free to reach out with any further questions! Thank you again for the valuable comments and feedback.
>
> [1] Damani, et al. "Beyond binary rewards: Training lms to reason about their uncertainty." arXiv:2507.16806 (2025).
>
> [2] Leang, et al. "Picsar: Probabilistic confidence selection and ranking for reasoning chains." arXiv:2508.21787 (2025).
>
> [3] Sutawika, et al. "EleutherAI/lm-evaluation-harness: v0. 4.9." Zenodo (2025).
>
> [4] Kojima, et al. "Large language models are zero-shot reasoners." Advances in neural information processing systems (2022).

---

> > ### Author Rebuttal · Reviewer_qgAU · 2026-04-01
> >
> > Thank you for the clarifications and additional experiments. The matched re-evaluation and added baselines address my concerns well and strengthen the evidence for DiffuSearch; overall, the response increases my confidence in the work. Nice work and good luck.

---

### Official Review · Reviewer_LjhA · 2026-03-12

**Soundness:** 4
**Presentation:** 3
**Significance:** 3
**Originality:** 3
**Overall Recommendation:** 5
**Confidence:** 4

**Summary:**

In the diffusion language model/masked diffusion model, the generation order is very important; the authors use MCTS to search for "which slot to fill next," thereby improving inference and code generation effects. They call this method MCDIFFUSE, and it is training-free, meaning that the model does not need to be retrained and only performs search during inference.

**Compliance With Llm Reviewing Policy:**

Affirmed.

**Final Justification:**

I thank the authors for their detailed response. I believe my current score remains appropriate.

**Key Questions For Authors:**

See weakness above.

**Limitations:**

yes

**Strengths And Weaknesses:**

### Strengths
1. MCDIFFUSE does not require retraining the model or an additional learned value model, but instead uses the existing model's confidence and rollout directly during inference for search.
2. Empirical results are persuasive.

### Weaknesses

1. Although the paper's claim is strong, its core experiments are based on the ReFusion base model, and the authors themselves note that, to their knowledge, it is "the only available slot-and-infill diffusion language model." This implies: It demonstrates that MCTS + ReFusion is effective, but does not sufficiently prove that MCTS slot planning is equally effective across more types of DLM/MDM.

2. The paper also acknowledges that MCTS brings additional overhead, and the generation time basically increases approximately linearly with the number of simulations. More importantly, the authors found that increasing the number of simulations from Nsim=30 to 270 typically results in an accuracy improvement of less than 2%, but the computational cost increases by 9 times.

But regardless, I personally think it is a good paper and worth accepting.

---

> ### Author Rebuttal · Authors · 2026-03-27
>
> We sincerely thank Reviewer LjhA for acknowledging the strength of our approach in leveraging MCTS with model confidence and rollout within slot-based infilling, achieving strong empirical performance without requiring any additional training. We also truly appreciate the reviewer's positive recognition and recommendation for acceptance!
>
> > **W1:** It demonstrates that MCTS + ReFusion is effective, but does not sufficiently prove that MCTS slot planning is equally effective across more types of DLM/MDM.
>
> We thank the reviewer for raising this concern. While our approach is built around slot-based infilling, and our evaluations therefore focus on ReFusion, we agree that extending MCTS slot planning to other DLM/MDM architectures remains an important direction. The primary challenge is that models such as Diffusion-LM and Block Diffusion operate at the token level guidance, making guided search more computationally expensive compared to slot-based guidance. However, we certainly agree that extending our approach to additional DLM/MDM architectures would be a key avenue for future work, and will discuss this explicitly in the camera-ready version.
>
> > **W2:** The authors found that increasing the number of simulations from Nsim=30 to 270 typically results in an accuracy improvement of less than 2%, but the computational cost increases by 9 times.
>
> We thank the reviewer for this observation. We acknowledge that increasing Nsim from 30 to 270 yields diminishing returns (less than 2% improvement at 9× the computational cost). However, while increasing simulations generally improves performance in MCTS, our analysis in Section 5.3 shows that leveraging broader exploration through CUPT is far more effective, achieving 13.6% performance gains at substantially lower cost. This is because increasing the exploration directs the search toward underexplored but promising reasoning paths without incurring the linear cost of additional rollouts. We will include a more detailed analysis of this compute-performance trade-off in the camera-ready version.
>
> Thank you again for the positive overall assessment and valuable feedback! We are happy to address any further concerns.

---

> > ### Author Rebuttal · Reviewer_LjhA · 2026-04-01
> >
> > I thank the authors for their detailed response. I believe my current score remains appropriate.
> >
> > Good luck!

---

### Official Review · Reviewer_WuE8 · 2026-03-16

**Soundness:** 2
**Presentation:** 2
**Significance:** 2
**Originality:** 2
**Overall Recommendation:** 4
**Confidence:** 4

**Summary:**

This paper applies MCTS to the slot-ordering problem in text diffusion decoding. Tokens are generated left-to-right within slots. Tree-search optimizes the average probability of a token within a slot. This results in a maximum-likelihood search procedure, with more flexible ordering than left-to-right. Experiments show gain over the ReFusion baseline results particularly in coding.

**Compliance With Llm Reviewing Policy:**

Affirmed.

**Key Questions For Authors:**

The only key question is: The reported coding scores for ReFusion seem lower than those reported in the original paper. Could you clarify this discrepancy? Compared to the ReFusion paper, only MBPP demonstrates gains.

Other comments and questions
* What is the definition of serial blocks in appendix C.1?
* Line 201 typo
* In Figure 1 and Section 3.2, how can Q(s,a) > 1?
* 5.3 over-claims coherence
* Page 8: "Together, These"

**Limitations:**

Yes

**Strengths And Weaknesses:**

Soundness: As of this review, this is the weakest component of the paper. The ReFusion baseline coding results are quite a bit lower than those reported in the ReFusion paper. Aside from the magnitude of the results, the analysis in the main paper and appendix are quite thorough.

Presentation: The paper was clearly written and well-structured. Since the method is an application of MCTS, I would recommend moving more of the analysis, especially the wall-clock time analysis, from the appendix to the main paper. The method in section 3 can be trimmed to be a single page of content.

Significance: The question of slot ordering is very relevant, and the analysis of mostly-sequential but impactful out-of-order is interesting.

Originality: The application of MCTS to slot-ordering is novel and interesting. I fear that, due to the clear left-to-right bias of reasoning and agentic trajectories, slots in text diffusion may not be the best application on the speed-accuracy frontier.

Overall, I believe this paper is a solid piece of scientific work with potentially negative results.

---

> ### Author Rebuttal · Authors · 2026-03-27
>
> We sincerely thank Reviewer WuE8 for recognising DiffuSearch as a solid piece of scientific work, and for acknowledging its novelty, interesting analysis, and well-written presentation. We would like to offer clarifications and provide additional results that may address most of the concerns.
>
> **Weaknesses**
> > **W1:** Clarification on the difference between our ReFusion baseline results and those reported in the original ReFusion paper.
>
> We thank the reviewer for raising this concern.  We will include the additional results reported below in the final version to clarify the discrepancy:
>
> We would like to clarify that the discrepancy in baseline results is due to differences in evaluation setup, such as the input prompts provided. Specifically, ReFusion adopts LM-Eval-Harness [1], which uses task-specific (i) prompts, (ii) answer extraction, and (iii) parsing logic. In contrast, our initial experiments used the standard CoT prompt [2] (i.e. “let's think step by step”). This was done intentionally to observe the reasoning traces and study the effect of introducing MCTS under a consistent prompting setup.
>
> We further re-evaluated all baselines using LM-Eval-Harness with matched decoding and configurations, and the improvements remain consistent.
>
> | Model | GSM8K | GPQA Diamond | MATH500 | MBPP | HumanEval | ARC_C |
> |---|---:|---:|---:|---:|---:|---:|
> | ReFusion | 84.91% | 32.32% | 54.00% | 68.20% | 78.66% | 89.76% |
> | DiffuSearch | **88.10%** | **36.87%** | **62.20%** | **78.40%** | **89.63%** | **91.98%** |
>
> Notably, coding benchmarks (MBPP, HumanEval) show the most substantial gains (>10%), while multiple-choice benchmarks (ARC-Challenge, GPQA) exhibit marginal improvements. MATH500 also demonstrates a meaningful gain of +8.2%, suggesting that MCTS is particularly effective on tasks requiring extended, multi-step reasoning. We will include the results for both experimental modes in the paper, highlighting the different settings.
>
> > **W2:** I would recommend moving more of the analysis, especially the wall-clock time analysis, from the appendix to the main paper.
>
> We thank the reviewer for this valuable suggestion! We agree that moving the wall-clock time analysis from the appendix to the main paper would strengthen the paper. We will include this in the camera-ready version.
>
> > **W3:** Slots in text diffusion may not be the best application on the speed-accuracy frontier.
>
> We thank the reviewer for the insightful comment. We agree that many reasoning and agentic trajectories exhibit a strong left-to-right bias, which can make slot-based diffusion less advantageous in some settings. However, strict left-to-right commitment can be overly restrictive when parts of the trajectory admit revision or alternative orderings (as supported by our ordering analysis). DiffuSearch is aimed at this setting, where search over partial refinements can allocate computation adaptively, favour decisions with higher long-term utility, and reduce the impact of early mistakes and potentially shortening trajectories. More broadly, we believe this is an interesting direction for future research, and we will revise the paper to make this scope more explicit.
>
> **Other comments and questions**
>
> > **Q1:** Definition of serial blocks in App. C.1?
>
> Based on ReFusion [3], we define serial blocks to refer to consecutive, non-overlapping sub-sequences that partition the target sequence. These blocks are generated in a serial, left-to-right order. In our experiments, we used a serial block size of 2, configured as half of the original serial block, which we will update and clarify in the camera-ready version.
>
> > **Q2, Q3, Q5:** Q(s,a) > 1?, Line 201 typo, Page 8: "Together, These"
>
> Thank you for bringing this to our attention. We acknowledge that Q(s,a) are bounded between [0,1]. We will correct the example values in Figure 1, update the serial block size to 2, and address all the mentioned typos.
>
> > **Q4:** 5.3 over-claims coherence
>
> While we initially used it to describe the model’s ability to maintain performance (as reported in Table 1) despite the shorter output lengths, we agree that we do not provide a separate, formal metric for coherence. To ensure technical clarity and avoid over-claiming, we have revised Section 5.3 to focus exclusively on compactness and its supporting quantitative evidence.
>
> We hope our responses address your concerns. If there are any further questions or suggestions, please let us know! Thank you again for the constructive feedback and recognition of our framework.
>
>
> [1] Sutawika, et al. "EleutherAI/lm-evaluation-harness: v0. 4.9." Zenodo (2025).
>
> [2] Kojima et al. "Large language models are zero-shot reasoners." Advances in neural information processing systems 35 (2022).
>
> [3] Li et al. "ReFusion: A Diffusion Large Language Model with Parallel Autoregressive Decoding." arXiv:2512.13586 (2025).

---

### Decision · Program_Chairs · 2026-04-30

**Decision:**

Accept (regular)

**Comment:**

All reviewers speak high of the contribution and vote for acceptance. They were impressed by the novel application of MCTS to masked diffusion inference, training-free nature of the method, and strong empirical results. There were concerns regarding baseline scores lower than reported in literature which is sufficiently addressed in rebuttal. Therefore, I'm recommending acceptance.

For the final version, the authors should revise the related work section carefully - I find them incomplete and the statement inaccurate. it is stated that "Masked Diffusion Models (MDMs). MDMs, originally successful in visual domains (Song et al., 2021; Ho et al.,
2020)". I think the statement here applies to diffusion models. Masked diffusion models was originally proposed in Austin et al. (2021) and successfully demonstrated to work only until recently - the correct references are

Shi, J., Han, K., Wang, Z., Doucet, A., & Titsias, M. (2024). Simplified and generalized masked diffusion for discrete data. Advances in neural information processing systems, 37, 103131-103167.

Sahoo, S. S., Arriola, M., Schiff, Y., Gokaslan, A., Marroquin, E., Chiu, J. T., ... & Kuleshov, V. (2024). Simple and effective masked diffusion language models. Advances in Neural Information Processing Systems, 37, 130136-130184.

Ou, J., Nie, S., Xue, K., Zhu, F., Sun, J., Li, Z., & Li, C. (2024). Your absorbing discrete diffusion secretly models the conditional distributions of clean data. arXiv preprint arXiv:2406.03736.